# EUGens: Efficient, Unified, and General Dense Layers

**Sang Min Kim**[*1]    **Byeongchan Kim**[*1]    **Arijit Sehanobish**[*2]    **Somnath Basu Roy Chowdhury**[*3]
**Rahul Kidambi**[*3]    **Dongseok Shim**[1]    **Avinava Dubey**[*3]    **Snigdha Chaturvedi**[4]
**Min-hwan Oh**[1]    **Krzysztof Choromanski**[*5,6]

[1]Seoul National University   [2]Independent   [3]Google Research   [4]UNC Chapel Hill

[5]Google DeepMind   [6]Columbia University

## Abstract

Efficient neural networks are essential for scaling machine learning models to real-time applications and resource-constrained environments. Fully-connected feedforward layers (FFLs) introduce computation and parameter count bottlenecks within neural network architectures. To address this challenge, in this work, we propose a new class of dense layers that generalize standard fully-connected feedforward layers, **E**fficient, **U**nified and **Gen**eral dense layers (EUGens). EUGens leverage random features to approximate standard FFLs and go beyond them by incorporating a direct dependence on the input norms in their computations. The proposed layers unify existing efficient FFL extensions and improve efficiency by reducing inference complexity from quadratic to linear time. They also lead to **the first** unbiased algorithms approximating FFLs with arbitrary polynomial activation functions. Furthermore, EuGens reduce the parameter count and computational overhead while preserving the expressive power and adaptability of FFLs. We also present a layer-wise knowledge transfer technique that bypasses backpropagation, enabling efficient adaptation of EUGens to pre-trained models. Empirically, we observe that integrating EUGens into Transformers and MLPs yields substantial improvements in inference speed (up to **27**%) and memory efficiency (up to **30**%) across a range of tasks, including image classification, language model pre-training, and 3D scene reconstruction. Overall, our results highlight the potential of EUGens for the scalable deployment of large-scale neural networks in real-world scenarios.

## 1   Introduction

Recent advances in machine learning (ML) have revolutionized the design and deployment of intelligent systems, with applications in robotics [8, 49], computer vision [40], natural language processing (NLP) [18, 91, 94, 1], and 3D scene understanding [61]. These advancements have been primarily driven by large-scale machine learning models, which are capable of solving increasingly challenging tasks. Despite the effectiveness of these models, computational inefficiencies often hinder their real-world deployment. Therefore, improving the efficiency of both training and inference of such models is essential to applying these methods in fast-paced, safety-critical domains.

A major source of computational cost in modern ML models is the use of fully connected feedforward layers (FFLs), which are central to many architectures. FFLs are integral to Transformers [97]—the backbone of Large Language Models (LLMs) and Vision Transformers (ViTs)—and to implicit neural representations (INRs) like Neural Radiance Fields (NeRF) [61] for 3D scene modeling. These layers account for a substantial portion of the model parameters and computation. In Transformers, fully connected feedforward layers (FFLs) are major contributors to the overall model size. Similarly,

---

[*]Equal Contribution

Correspondence to: `tkdals9082@snu.ac.kr`, `arijit.sehanobish1@gmail.com`, `kchoro@google.com`

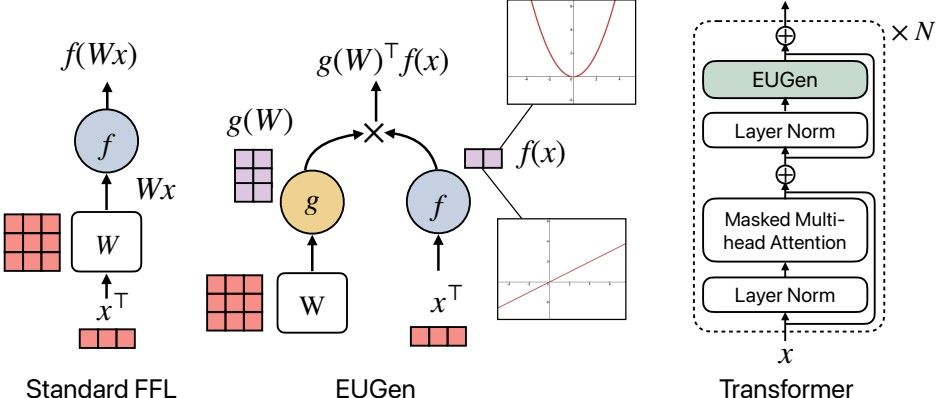

Figure 1: Schematic diagram showcasing the workflow of a standard fully connected (*left*) and EUGen layer (*middle*). In EUGen, both the input, $\mathbf{x}$, and weight, $\mathbf{W}$, are transformed by non-linear mapping $f$ and $g$. These operations produce low-dimensional matrices whose multiplication reduces computational cost. Different dimensions of the new representation of the input $\mathbf{x}$ correspond to different monomials in the polynomial approximation of the activation function $f$. The representation can also directly depend on $\|\mathbf{x}\|_2$ (see: Sec. 3). Such EUGen layers are introduced in Transformer blocks (*right*) to improve the efficiency of the overall network.

architectures like NeRF and iSDF [69] use FFLs to encode high-dimensional spatial representations, supporting real-time tasks such as photorealistic rendering. Therefore, reducing the computational complexity of FFLs is essential for enabling the practical deployment of these architectures.

In this work, we present a new class of efficient dense layers that generalize their regular fully-connected feedforward counterparts (FFLs), unify other proposed extensions, and provide efficiency by replacing quadratic- with linear-time inference. These **E**fficient, **U**nified, and **Gen**eral dense layers (*EUGens*) can be seamlessly integrated into implicit neural representations and Transformer architectures, significantly reducing computational complexity and parameter count without sacrificing expressivity. The key innovation of EUGens lies in their ability to disentangle the processing of weights and inputs, connecting them through simple linear operations (as shown in Fig. 1). Inspired by the seminal work of Rahimi and Recht [73] on random features (RFs), EUGens employ RF transformations to process the inputs. They effectively act as kernel methods, enabling efficient approximation of complex interactions in high-dimensional data. EUGens can approximate regular FFLs and even go beyond them by introducing direct dependence on the inputs' norms.

By incorporating EUGens into Transformers and NeRFs, we show substantial improvements in inference speed (up to **27**%) and memory usage (up to **30**%) across diverse tasks: image classification, NLP, and 3D scene reconstruction. We also propose a layer-wise knowledge transfer technique that bypasses backpropagation, enabling efficient adaptation of EUGens into pre-trained models. Our results highlight the potential of EUGens for scalable deployment in real-world scenarios. To summarize, our main contributions are as follows:

- We propose a new class of dense layers, EUGens, which leverages random features to approximate feedforward layers and provide unbiased estimation for polynomial activation functions (Section 3).
- We present an extensive theoretical analysis of EUGen's estimation with the concentration bounds (Theorem 3.2 and 3.3) and propose additional improvements via QMC techniques (Section 3.1).
- We empirically show that EUGens accurately approximate FFLs with common activations (e.g., ReLU, GeLU, Softplus), significantly outperforming baseline methods (Section 4.1).
- We integrate EUGens into Transformers and NeRFs as FFL replacements, enabling faster deployment across diverse NLP, vision, and 3D scene modeling tasks (Section 4.2, 4.3, and 4.4).
- We propose a distillation framework that replaces existing feedforward layers with EUGens, which can be trained analytically *without backpropagation*. This approach enables zero-shot integration with pretrained models, improving inference speed without retraining (Section 4.5).

The proofs of the theorems (Theorem 3.1 introducing EUGens that give unbiased estimations for the FFLs with polynomial activations, Theorem 3.2 and Theorem 3.3 providing concentration results and Theorem 3.4 presenting extension to general continuous activations) are given in the Appendix.

## 2 Related Work

A significant challenge in training neural networks is the quadratic scaling of their space and time complexity with the size of the hidden layers. To address this challenge, many works have focused on reducing these computational bottlenecks. One of the popular ways to tackle this issue is via dimensionality reduction à la Johnson-Lindenstrauss Transform (or JLT) [22, 21, 2] or via kernel methods with random features (RF) [73]. Since the influential work of Rahimi and Recht [73], substantial effort has been dedicated to improving the accuracy of RF-based methods by introducing entangled projections [119, 17, 16]. Notably, kernels of the form $\mathbf{K}_f(\mathbf{x}, \mathbf{y}) := f(\mathbf{x}^\top \mathbf{y})$ where $f :=$ $\exp$ [15, 53] or $f$ is a polynomial or power series with positive coefficients [47, 100, 101, 99, 3, 46] have been extensively studied. However, these approaches are limited to positive definite kernels with only a subset [15, 53, 3] targeting the attention module in the Transformer.

Kernel methods have been used to linearize two-layer neural networks [13, 14] and to analyze general networks using Neural Tangent Kernels (NTK) [44] and Path Kernels [26]. The popular NTK-based methods are suitable for explaining neural network training during the fine-tuning phase when the weights do not deviate much from the pre-trained weights [107, 60, 92]. Therefore, none of these methods can be applied directly to reduce the complexity of FFL layers when training from scratch. Closest to our approach, Sehanobish et al. [80] proposed the Universal Random Feature (URF), which disentangles weights and inputs in specific MLP layers. However, their method relies on Fourier transforms of activation functions, limiting its applicability to commonly used activation functions in machine learning. In particular, it cannot be applied to unbounded activation functions (e.g. ReLU).

Beyond kernel methods, common approaches to reduce the computational complexity of dense-layer models are via pruning [85, 59], quantization [111, 57, 124], knowledge distillation [51, 83, 9], or replacing dense matrices with *structured ones* [70, 108] – all widely used in Transformers. In the context of NeRFs, various specialized methods have been proposed to enhance their efficiency [89, 112, 84, 75, 64, 42, 76]. Similarly, techniques tailored to Signed Distance Fields (SDF) architectures have been developed to enable their application in real-world scenarios [69, 72, 103, 106].

In contrast to the above solutions, EUGen layers provide a general-purpose mechanism applicable across diverse architectures, including Transformers, NeRFs, and incremental SDF models. The formulation of EUGens also allows post-training compression and fast distillation using a closed-form solution. Finally, we would like to highlight that EUGens is orthogonal to common efficiency techniques like sparse sampling, pruning, etc., and can be combined with such to further reduce the computational footprint. For completeness, we discuss additional related techniques in Appendix C.

## 3 Efficient, Unified, and General Dense Layers (EUGens)

In this section, we formally define EUGen layers. Take a weight matrix row $\mathbf{w} \in \mathbb{R}^d$ (in the column format) and an input vector $\mathbf{x} \in \mathbb{R}^d$. The $k^{th}$-order EUGen layer $\text{EUGen}^k(\mathbf{w}, \mathbf{x})$ is given as follows:

$$\text{EUGen}^k(\mathbf{w}, \mathbf{x}) = \left\langle \Psi \left( \text{concat} \left( \prod_{j=1}^{i} \mathbf{G}_j^i \mathbf{w}^+ \right)_{i=0,...,k} \right), \Phi \left( \text{concat} \left( \prod_{j=1}^{i} \mathbf{G}_j^i \mathbf{x}^+ \right)_{i=0,...,k} \right) \right\rangle \tag{1}$$

Here $\mathbf{w}^+$ and $\mathbf{x}^+$ are obtained from $\mathbf{w}$ and $\mathbf{x}$ by concatenating with 1 and $\|\mathbf{x}\|_2$ respectively, $\langle \rangle$ stands for the dot-product, $\prod$ defines Hadamard-product (i.e. element-wise multiplication), concat denotes concatenation of the vectors, $\mathbf{G}_j^i \in \mathbb{C}^{m \times (d+1)}$ (for $i = 0, ..., k$, $j = 1, ..., i$), and $\Psi, \Phi : \mathbb{C} \to \mathbb{R}$ ($\Psi, \Phi$ are applied element-wise). For the weight matrix $\mathbf{W} \in \mathbb{R}^{l \times d}$, the EUGen layer of $k^{th}$ order $\text{EUGen}^k(\mathbf{W}, \mathbf{x})$ is given by concatenating values $\text{EUGen}^k(\mathbf{w}, \mathbf{x})$ for different rows $\mathbf{w}$ of $\mathbf{W}$.

Intuitively speaking, order $k$ controls the expressivenes of the layer. Indeed, as we will see later (Theorem 3.1), even for identity functions $\Psi$ and $\Phi$, EUGens are capable of accurately approximating fully-connected feedforward layers (FFLs) with complex activation functions for the appropriately tuned hyperparameter $k$.

Note that EUGen layers disentangle $\mathbf{W}$ from $\mathbf{x}$, only to connect them in the final computations of the dot-product (linear) kernels (shown in Fig. 1). Indeed, $\text{EUGen}^k(\mathbf{w}, \mathbf{x})$ can be re-written as:

$\text{EUGen}^k(\mathbf{w}, \mathbf{x}) = g(\mathbf{w})^\top f(\mathbf{x})$, where functions $f, g$ are defined as follows:

$$f(\mathbf{x}) = \Phi\left(\text{concat}\left(\prod_{j=1}^{i} \mathbf{G}_j^i \mathbf{x}^+\right)_{i=0,\dots,k}\right), g(\mathbf{w}) = \Psi\left(\text{concat}\left(\prod_{j=1}^{i} \mathbf{G}_j^i \mathbf{w}^+\right)_{i=0,\dots,k}\right) \quad (2)$$

**Inference time complexity:** Here only matrix-vector products $\mathbf{G}_j^i \mathbf{x}^+$ need to be computed (for the pre-computed $g(\mathbf{w})$). Under an assumption that applying $\Phi$ per-entry takes constant-time, inference can be conducted in time $O(mdk^2 + ml)$. In practice we use small $k$ ($\leq 3$), thus for $m \ll \min(d, l)$ significant gains can be obtained, as compared to the brute-force $O(d^2 + dl)$ variant, since time complexity becomes linear in $d$. We demonstrate it in Sec. 4.2, 4.3, and 4.4. If we drop dependence on $i$ in the definition of $\mathbf{G}_j^i$ (i.e. $\mathbf{G}_j^i = \mathbf{G}_j$), time complexity can be further improved to $O(mdk + ml)$.

We are ready to formulate our main theoretical result. In the theorem below, we show that EUGens are capable of unbiasedly approximating FFLs with **arbitrary** polynomial activation functions $f$ and with easy to construct matrices $\mathbf{G}_j^i$. To the best of our knowledge, this is the first such result. The most relevant previous results from [80] assumed the existence of the *Fourier Transform* (FT) of $f$.

---

**Theorem 3.1** (EUGens can unbiasedly approximate FFLs with polynomial activations). *Take the* FFL *defined as follows for a weight matrix* $\mathbf{W} \in \mathbb{R}^{l \times d}$ *and an input vector* $\mathbf{x} \in \mathbb{R}^d$: $\text{FFL}(\mathbf{W}, \mathbf{x}) = f(\mathbf{W}\mathbf{x})$ *(note that this is the most general form, since the bias term can always be absorbed by adding extra column to matrix* $\mathbf{W}$ *and extra entry to input* $\mathbf{x}$*), where the activation function* $f : \mathbb{R} \to \mathbb{R}$ *is given as:*

$$f(x) = \sum_{i=0}^{k} a_i x^i. \quad (3)$$

*Choose* $\Psi, \Phi$ *as identity functions. Take some zero-mean distributions:* $\mathcal{D}_j^i \in \mathcal{P}(\mathbb{R})$ *for* $i = 0, \dots, k$ *and* $j = 1, \dots, i$*, with standard deviations* $\sigma_{i,j}$*. Assume that the last columns of the matrices* $\mathbf{G}_j^i$ *are all zero and entries of the other columns of matrices* $\mathbf{G}_j^i$ *are sampled independently as:*

$$\mathbf{G}_j^i(\cdot, \cdot) \sim \frac{1}{\sigma_{i,j} m^{\frac{1}{2i}} |a_i|^{\frac{1}{2i}} \xi_i(2i)} \mathcal{D}_j^i, \quad (4)$$

*where* $\xi_i(t) \in \mathbb{C}$ *satisfies:* $\xi_i^t(t) = sgn(a_i)$*. Then the following holds:*

$$\mathbb{E}\left[\text{EUGen}(\mathbf{W}, \mathbf{x})\right] = f(\mathbf{W}\mathbf{x}) \quad (5)$$

*The results holds also if* $\mathbf{G}_j^i$ *does not depend on* $i$*, i.e.* $\mathbf{G}_j^i = \mathbf{G}_j$*, as well as when rows of matrices* $\mathbf{G}_j^i$ *are dependent, as long as entries within each row are chosen independently.*

---

By setting the last columns of the matrices $\mathbf{G}_j^i$ as zeros in Theorem 3.1, we do not leverage the opportunity, that EUGens give, to provide a direct dependence on $\|\mathbf{x}\|_2$. That suffices to unbiasedly and accurately approximate FFLs with polynomial activation functions (and in practice any continuous $f$ since, by the Bernstein Theorem [78], any continuous function defined on a closed interval can be approximated with arbitrary accuracy by a polynomial). Adding direct dependence on $\|\mathbf{x}\|_2$ enables EUGens to provide computational models beyond the scope of regular FFLs.

Concentration results for the approximation mechanism presented in Theorem 3.1, in the form of the variance formula, are given in Theorem 3.2. Additional results, providing exponentially small probabilities of large divergence from the mean and leveraging Azuma's inequality, are given in Theorem 3.3. We complement our theoretical analysis with the result showing how EUGens with polynomial activation functions can be used to approximate FFLs with continuous activations (Theorem 3.4). Those results leverage polynomial approximation of the continuous functions.

The sketch of the proof of Theorem 3.3 is as follows. We observe that the EUGens are effectively conducting an unbiased Monte Carlo approximation of the output values of the regular FFL. Under the assumptions of the theorem, this is achieved by averaging over independent and bounded random variables. We can then apply standard concentration inequalities, such as Azuma's Inequality.

**Theorem 3.2** (Concentration results of EUGens: part I). *Under the assumption from Theorem 3.1 (the entries of all $\mathbf{G}_j^i$ for $i = 0, ..., k$, $j = 1, ..., i$ sampled independently), the variance of the estimation $Z = \widehat{\mathrm{FFL}}(\mathbf{W}, \mathbf{x})[u]$ of $\mathrm{FFL}(\mathbf{W}, \mathbf{x})[u]$ for $u = 1, ..., l$ is given as follows for $\tau_{i,j}$ denoting the fourth moment of the random variable sampled from $\frac{1}{\sigma_{i,j}}\mathcal{D}_j^i(\cdot, \cdot)$, $\mathbf{w}(u)$ denoting the $u^{th}$ row of $\mathbf{W}$ (in the column format) and $\rho_i = (\mathbf{w}(u)^\top \mathbf{x})^{2i}$:*

$$\mathrm{Var}(Z) = \frac{1}{m} \sum_{i=0}^{k} \left( \left( 2(\mathbf{w}(u)^\top \mathbf{x})^2 + \|\mathbf{w}(u)\|_2^2 \|\mathbf{x}\|_2^2 + (\tau_{i,j} - 3) \sum_{s=1}^{d} \mathbf{w}(u)_s^2 x_s^2 \right)^i - \rho_i \right) a_i^2 \tag{6}$$

**Theorem 3.3** (Concentration results of EUGens: part II). *Consider the setting from Theorem 3.2 and assume furthermore that the absolute values of the entries of random variables taken from $m^{\frac{1}{2i}}\mathcal{D}_j^i$ are upper-bounded by $c$ for some $c > 0$. Then the following holds for $\eta_1 = \max_{i=0,...,k} \left| a_i (\mathbf{w}(u)^\top \mathbf{x})^i \right|$, $\eta_2 = \max_{i=0,...,k} \left| a_i (c^2 d \|\mathbf{x}\|_2 \|\mathbf{w}(u)\|_2)^i \right|$ and any $\epsilon > 0$:*

$$\mathbb{P}[|\widehat{\mathrm{FFL}}(\mathbf{W}, \mathbf{x})[u] - \mathrm{FFL}(\mathbf{W}, \mathbf{x})[u]| \geq \epsilon] \leq h(\epsilon) \stackrel{\mathrm{def}}{=} 2 \exp\left(-\frac{m\epsilon^2}{2(\eta_1 + \eta_2)^2 k^2}\right) \tag{7}$$

**Remark I:** The upper bounds on failure probabilities in Theorem 3.3 and Theorem 3.4 are exponentially small in the number of random features $m$. Consequently, strong guarantees regarding accurate approximation in several points simultaneously can be directly obtained via the union bound trick.

**Remark II:** In practice, EUGens are often initialized to provide an unbiased approximation of the well-defined regular FFL layers, but then matrices $\mathbf{G}_j^i$ are usually trained.

**Remark III:** Although pure polynomial-activation networks (PNNs) have limited expressiveness unless the degree $k$ is large, this limitation does not directly apply when polynomial layers are combined with standard activation layers. Our models use hybrid architectures that mix EUGen layers with regular feedforward layers, so the PNN theory no longer applies. Empirically, across multiple modalities, small-degree hybrids match the performance of standard models while offering substantial computational benefits (see Secs. 4.2, 4.3, and 4.4). Accordingly, our approximation results should be interpreted in the hybrid setting rather than in the context of pure PNNs.

**Remark IV:** Since the EUGen layers disentangle the weights and the inputs, they can be trivially combined with a subsequent linear layer. This enables us to compress models like Transformers and NeRFs where an EuGen layer is naturally followed by linear layers (more details presented in Fig. 11 and in Sec. B.1).

## 3.1 Quasi Monte Carlo (QMC) EUGens

In this section, we assume that $m \leq d$ (which has to be the case anyway for computational gains). Since in this section, as in Theorem 3.1, the last columns of $\mathbf{G}_j^i$ will be zeroed, without loss of generality we will assume that: $\mathbf{G}_j^i \in \mathbb{R}^{d \times d}$. The standard initial choices for the matrices $\mathbf{G}_j^i$ are Gaussian matrices. Note that for the special case of $k = 1$, EUGens effectively conduct JLT dimensionality reduction with projection matrices $\mathbf{G}_j^i$. It is a well-known fact that structured random matrices with correlated rows can reduce the variance of the JLT estimation of the dot-product kernel. In particular, *Gaussian orthogonal matrices* (GOMs) [119] from $\mathbb{R}^{d \times d}$ (with Gaussian marginal distributions of rows, but exactly orthogonal rows), truncated to their first $m$ rows **provably** reduce that variance [17]. We applied this approach to EUGens with general $k$ ($k > 1$) and empirically confirm its effectiveness, also in the setting with EUGens directly depending on $\|\mathbf{x}\|_2$ (see: Sec. 4.1).

**Theorem 3.4** (EUGens approximating FFLs with general continuous activations). *Take an FFL using continuous activation $f$ with the associated modulus of continuity function $\omega$, defined as:*

$$\omega(x) = \sup_{|t_1 - t_2| \leq x} |f(t_1) - f(t_2)| \tag{8}$$

*Assume that $|\mathbf{w}(u)^\top \mathbf{x}| \leq \xi$ for some $\xi > 0$. Then there exists $C(\xi) > 0$ such that the following is true. For any $k > 0$ there exists a $k^{th}$-order EUGen layer such that for any given $\mathbf{x}$, $\epsilon > 0$ and $u \in \{1, ..., l\}$, the probability that $|\widehat{\mathrm{FFL}}(\mathbf{W}, \mathbf{x})[u] - \mathrm{FFL}(\mathbf{W}, \mathbf{x})[u]| \geq \epsilon + C(\xi)\omega(\frac{1}{\sqrt{k}})$ is upper-bounded by $p_{\mathrm{bound}} = \min(p_1, p_2)$, where (for $h(\cdot)$ as in Theorem 3.3):*

$$p_1 = \frac{\mathrm{Var}\left(\widehat{\mathrm{FFL}}(\mathbf{W}, \mathbf{x})[u]\right)}{\epsilon^2}, p_2 = h(\epsilon). \tag{9}$$

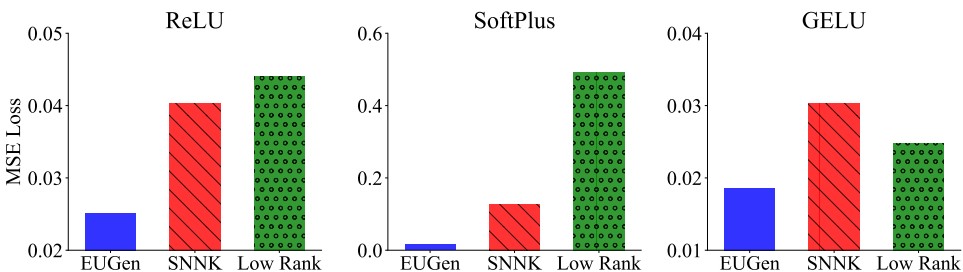

Figure 2: Approximation capability of our EUGen layer using (*left*) ReLU, (*middle*) Softplus and (*right*) GELU activation functions. EUGens provide superior approximation results compared to the baselines (SNNK and Low-Rank methods) with the same number of parameters.

# 4 Experiments

In this section, we outline the experimental setup and evaluate the performance of EUGen. Specifically, we design experiments to answer the following research questions:

(**RQ1**) *How effective are EUGens in approximating fully-connected feedforward layers (FFLs)?*

(**RQ2**) *How well do large-scale neural networks (e.g., Transformers & INRs) with EUGens perform?*

(**RQ3**) *How much speedup do EUGens achieve in large-scale neural networks?*

Next, we answer those research questions by evaluating EUGens in a variety of architectures, such as Transformers (e.g., LLMs and ViTs) and NeRFs. We applied EUGens of order $k \leq 2$ in all settings (further ablations on higher $k$ can be found in Appendix E.6.) We provide more details about our experimental setup in Appendix D.

## 4.1 Approximation Quality of EUGen in Synthetic Settings

In this experiment, we evaluate the approximation quality of EUGen layers. Specifically, we test whether EUGen layers can approximate the outputs of a fully-connected feedforward layer (FFL), $\mathbf{Y} = f(\mathbf{W}\mathbf{x} + \mathbf{b})$, where $f$ is an activation function. We compare EUGen's performance with low-rank approximation and SNNK [80] a special case of EUGen, for three different activation functions (ReLU, SoftPlus, and GELU). We report the results in Fig. 2, where we compute the MSE loss between the FFL output, $\mathbf{Y}$, and EUGen prediction, $\hat{\mathbf{Y}}$. We observe that EUGens achieve significantly lower MSE loss across all activations, showcasing a superior approximation quality of FFLs. This provides the answer to (**RQ1**), showing that EUGens are effective in approximating FFLs. We conducted additional experiments in synthetic setups (see Appendix D.1 for more details).

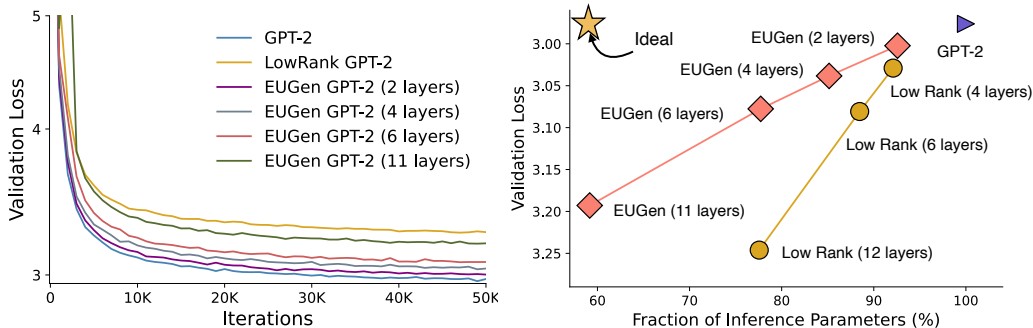

Figure 3: Evaluation result of EUGen for language model pre-training using GPT-2 (86M parameters). (*Left*) We report the validation loss of GPT-2 with different numbers of EUGen layers during pre-training. (*Right*) Tradeoff plot between the number of inference parameters and validation loss. Overall, we observe that EUGens outperform LowRank variations while enabling significant speedups with minimal impact on validation loss.

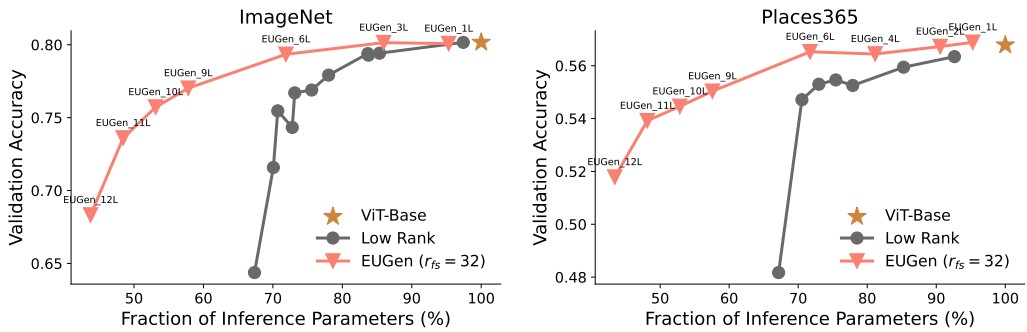

Figure 4: Evaluation results of EUGen for image classification tasks: (*left*) ImageNet and (*right*) Places365 datasets using ViT$_{base}$ (86M parameters). We observe that EUGens can match the performance of vanilla ViTs with a significantly smaller fraction of parameters than the Low-Rank baseline.

## 4.2 EUGens in LLM Pre-training

In this experiment, we evaluate the performance of EUGen in the context of LLM pre-training. We consider the GPT-2 Transformer architecture (with 124M parameters) and replace the FFLs with EUGen layers. We pre-train this architecture on ∼36.8B tokens from OpenWebText [35] dataset over 50K iterations. We compare our approach against the default GPT-2, GPT-2 with FFLs replaced by low-rank matrices, and baselines with varying numbers of EUGen layers.

We report the pre-training results in Fig. 3 (left). We observe that EUGens are good approximations of FFLs, achieving validation loss similar to vanilla GPT-2. We also notice that increasing the number of EUGen layers slightly increases the loss, which is expected due to the error accumulation effect. In Fig. 3 (right), we report the trade-off between the validation loss and the number of inference parameters (as a fraction of the vanilla GPT parameters). EUGens achieve good validation loss (*y*-axis is reversed) while using significantly fewer parameters. This provides the answer to **RQ2** & **RQ3**, showing that EUGens, in the context of LLM pre-training, achieve good performance using a significantly smaller number of parameters.

## 4.3 EUGens in Vision Transformers

In this experiment, we evaluate the efficacy of EUGens in Vision Transformers (ViTs) [29]. Similar to the last setting (Section 4.2), we replace fully connected feedforward layers (FFLs) with EUGens and use the resultant ViT architecture for image classification. In this setting, we systematically replace all the FFLs with EUGen and evaluate it on the ImageNet [23] and Places365 [125] datasets.

In Figure 4, we report the accuracy vs inference parameter tradeoff of EUGen and baseline methods. We observe that EUGens achieve the best trade-off curve, closely matching the performance of the vanilla ViT while using significantly fewer parameters (e.g., replacing six layers reduces the number of inference parameters by nearly **30**%). This provides further evidence to answer **RQ2** & **RQ3**, showcasing that EUGens are effective in achieving good image classification performance while using a fraction of trainable parameters. Additional results for image classification (including results with ViT-L) and language processing with Transformers are presented in Appendix D.3 and D.4.

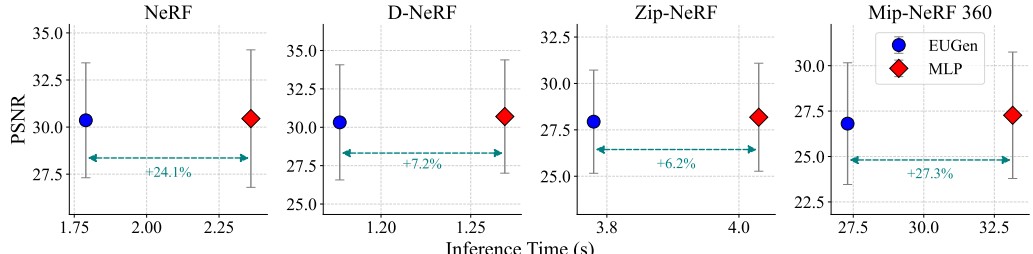

Figure 5: Quantitative results for NeRF experiments including NeRF, D-NeRF, Zip-NeRF, and Mip-NeRF 360 showing PSNR versus inference time. Our models achieve similar PSNR scores while achieving at least **24**% improvement in speeds for implicit representation models (NeRF and Mip-NeRF) as well as speeding the efficient hybrid models D-NeRF and Zip-NeRF by at least **6**%.

## 4.4 EUGens for Neural 3D Reconstruction

In this section, we show how EUGens can be seamlessly injected into various neural 3D scene reconstruction methods including NeRF [61] with demonstrated photo-realistic scene generation capability, Mip-NeRF 360 [4] the state-of-the-art (SOTA) architecture for novel video synthesis, Zip-NeRF [5], D-NeRF [71] and iSDF [69], a real-time module designed to reconstruct SDF from depth sequences. In all cases, we replaced up to three layers with EUGens (details in Appendix D). We applied EUGens explicitly using input's length $\|\mathbf{x}\|_2$ (see: Sec. 3).

In Fig. 5, we present the quantitative results of this replacement, plotting PSNR with inference time. For NeRFs [61], EUGen significantly reduces both inference time by **24**% (Fig. 5 (left)) and model size by **30**% (Tab. 8), respectively, with virtually no loss in reconstruction quality. For Mip-NeRF, on 360 V2 dataset [4] we achieve an **27**% (Fig. 5 (right))increase in inference speed while maintaining a similar reconstruction quality. We show qualitative results in Fig. 6.

For iSDF, in Fig. 7, we compare the original iSDF with EUGen replacing MLP layers (see Fig. 21 for all results and Table 4 for quantitative results). EUGens achieve similar performance to iSDF while being **22.6**% faster during inference and achieving a **5**% improvement in training speed. When we visualize slices of the reconstructed SDFs on the xy-plane at $z = 70$ cm, the reconstruction quality remains quite similar. Similar to the previous section, these results answer questions: **RQ2** and **RQ3**.

## 4.5 Knowledge Distillation with EUGens

In the previous sections, we showed that EUGen layers can significantly accelerate the inference time of implicit neural representations. However, achieving this acceleration typically requires retraining the model from scratch, which is computationally expensive and time-consuming. In this section, we show results on directly replacing the FFL layers with simple layer-wise distillation. This approach eliminates the need for per-scene retraining, enabling the acceleration of existing pretrained models without the overhead of rebuilding them from scratch. By storing the inputs and outputs of the target hidden layer in trained implicit neural representations and then optimizing the EUGen layer using a mean squared error objective, we can efficiently replicate the behavior of the original model.

This optimization has a closed-form solution and can be performed without *backpropagation* when $\mathbf{G}_j^i$ are sampled from a fixed distribution. We will refer to this EUGen variant as the 'Analytic' variant (see Appendix B.2). For NeRF, we show that we can recover the quality of the original model while improving inference speeds by up to **26**% (Fig. 8 and Table 5 in Appendix E.4).

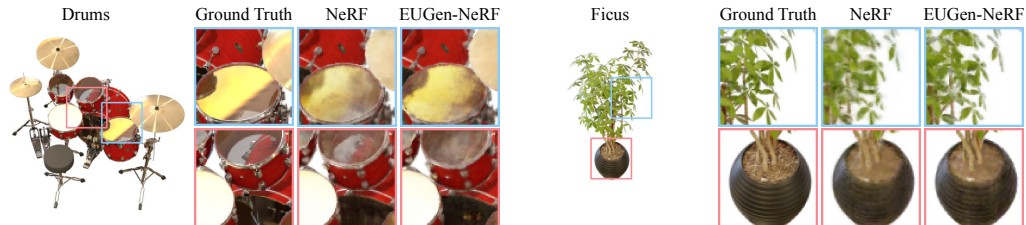

Figure 6: Results of rendering through NeRF versus EUGen-NeRF: *drums* (left) and *ficus* (right). The Ground Truth column shows the reference image, followed by the rendered results. We observe that the EUGen-NeRF renderings are indistinguishable from the NeRF renderings.

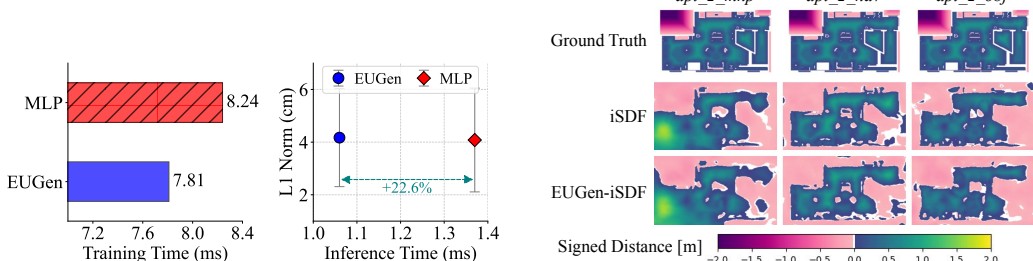

Figure 7: (Left) Training and inference comparisons between iSDF and EUGen-iSDF. EUGen-iSDF achieves **5**% faster training and **23**% faster inference with comparable reconstruction accuracy. (Right) Visualizations of the SDF reconstructions on ReplicaCAD scenes using iSDF (colormap below).

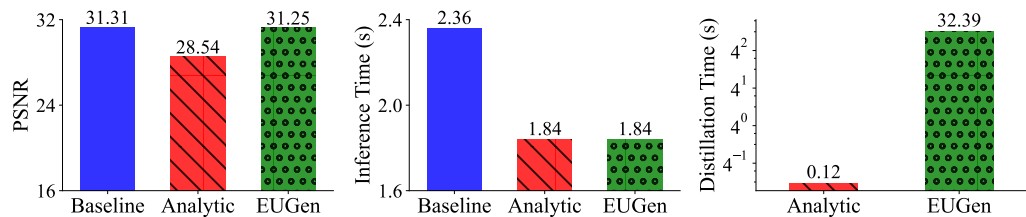

Figure 8: Distillation results for NeRF comparing the analytic and our EUGen variant with the baseline. (*Left*): Comparing PSNR of the baseline model with the distilled versions. (*Middle*): Comparing inference speed of baseline vs our distilled models. (*Right*): Comparing the distillation speed of analytic vs the EUGen variant.

## 4.6 Ablation Studies

In this section, we analyze how EUGens' key design choices impact performance. Specifically, we investigate the impact of trainable projection matrices ($\mathbf{G}_j^i$), the number of random features, and compare EUGens with dimensionality reduction baselines. See Appendix E for detailed ablations.

**Trainable Projection Matrices Improve Performance.** We study the impact of trainable vs. fixed $\mathbf{G}_j^i$ matrices in the iSDF setting, measuring reconstruction accuracy and speed. Consistent with prior findings [19, 121]), we observe that trainable $\mathbf{G}_j^i$ matrices enhances performance across all downstream tasks as shown in Fig. 9 (left & middle).

**Number of Random Features ($m$).** Fig. 9 (right) reveals the trade-off between computational speed and reconstruction quality with increasing number of random features. We find that increasing the random features improves performance (lower distance) while increasing inference time. We perform additional RF-based ablations in Appendix E.2 (see Table 8, 10, Fig. 26, 27).

**EUGen Models vs. Dimensionality Reduction.** Reducing hidden dimensions ($d$) is a common approach to accelerate training. To ensure a fair comparison, we decrease the hidden channels of

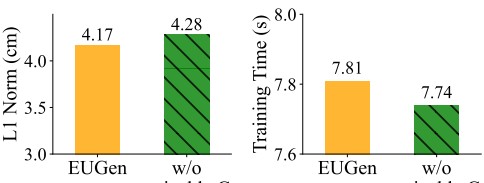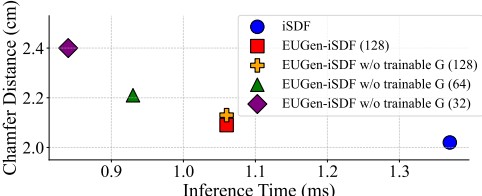

Figure 9: (*Left* & *Middle*) Comparing the performance of EUGen-iSDF with or without trainable $\mathbf{G}_j^i$. The non-trainable $\mathbf{G}_j^i$ is faster in training but shows slightly worse performance. (*Right*) Trade-off between reconstruction quality and speed for EUGen-iSDF models with 32, 64, 128 random features.

baseline models to match the training parameters of their EUGen counterparts. As shown in Fig. 10, the EUGen models consistently outperform these dimensionality reduction baseline models.

## 5    Conclusion

We introduce a novel type of neural network layer, termed EUGen, which leverages random features to efficiently approximate the computations of fully connected feedforward layers. By integrating EUGen into LLMs, Transformers, Neural Radiance Fields (NeRF), and Signed Distance Fields (SDF) models, we achieve a reduction in training parameters while preserving expressivity. Our approach enhances inference speed across all downstream applications for a wide range of model architectures. Additionally, the straightforward design of EUGens facilitates layer-wise

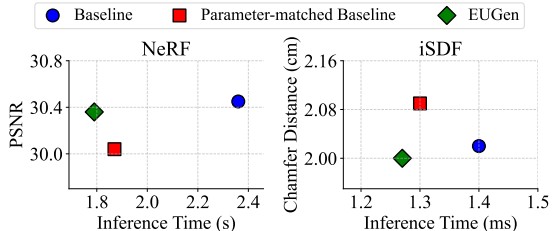

Figure 10: We compare the performance of EUGen models with baselines using (*left*) NeRF and (*right*) iSDF architectures. In both settings, EUGen models achieve better reconstruction quality than the baselines with the same parameter count while being much faster.

knowledge distillation without requiring backpropagation. We demonstrate that this distillation process leads to efficient inference. Overall, EUGens are a key contribution towards improving the efficiency of large-scale machine learning systems without impacting their expressivity.

## Acknowledgements

BK and MO were supported by the National Research Foundation of Korea (NRF) grant funded by the Korea government(MSIT) (No. RS-2022-NR071853 and RS-2023- 00222663).

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

# Appendix for EUGens: Efficient, Unified, and General Dense Layers

## A   Proofs of the Theoretical Results

In this section, we present the proofs of all our theoretical results.

### A.1   Proof of Theorem 3.1

*Proof.* Note that, by the definition of the EUGen layer, we have the following (under the conditions of the theorem) for $\mathbf{g}_j^i \in \mathbb{R}^{m \times d}$ with mean-zero entries of variance one each, and where $\{\mathbf{g}_j^i[r][l]\}_{l=1,...,d}$ is an independent set of random variables for every $r = 1, ..., m$:

$$\text{EUGen}(\mathbf{w}, \mathbf{x}) = \sum_{i=0}^{k} a_i \frac{1}{m} \sum_{r=1}^{m} \prod_{j=1}^{i} \left( \sum_{l=1}^{d} \mathbf{g}_j^i[r][l]\mathbf{x}_l \right) \left( \sum_{l=1}^{d} \mathbf{g}_j^i[r][l]\mathbf{w}_l \right) \tag{10}$$

Thus for any $\mathbf{w} \in \mathbb{R}^d$, we get :

$$\mathbb{E}[\text{EUGen}(\mathbf{w}, \mathbf{x})] = \sum_{i=0}^{k} a_i \frac{1}{m} \sum_{r=1}^{m} \mathbb{E}\left[ \prod_{j=1}^{i} \left( \sum_{l=1}^{d} \mathbf{g}_j^i[r][l]\mathbf{x}_l \right) \left( \sum_{l=1}^{d} \mathbf{g}_j^i[r][l]\mathbf{w}_l \right) \right] \tag{11}$$

Since $\{\mathbf{g}_j^i\}_{j=1}^{i}$ is an independent set of matrices for any $i = 0, ..., k$, we can re-write:

$$\mathbb{E}[\text{EUGen}(\mathbf{w}, \mathbf{x})] = \sum_{i=0}^{k} a_i \frac{1}{m} \sum_{r=1}^{m} \prod_{j=1}^{i} \mathbb{E}\left[ \left( \sum_{l=1}^{d} \mathbf{g}_j^i[r][l]\mathbf{x}_l \right) \left( \sum_{l=1}^{d} \mathbf{g}_j^i[r][l]\mathbf{w}_l \right) \right] \tag{12}$$

We will now prove the following lemma:

**Lemma A.1.** *The following is true:*

$$\mathbb{E}\left[ \left( \sum_{l=1}^{d} \mathbf{g}_j^i[r][l]\mathbf{x}_l \right) \left( \sum_{l=1}^{d} \mathbf{g}_j^i[r][l]\mathbf{w}_l \right) \right] = \mathbf{w}^\top \mathbf{x} \tag{13}$$

*Proof.* The proof of the above is in fact completely analogous to the proof of the unbiasedness of the regular *Johnson-Lindenstrauss Transform* (JLT), but we provide it again here for completeness. We have:

$$\mathbb{E}\left[ \left( \sum_{l=1}^{d} \mathbf{g}_j^i[r][l]\mathbf{x}_l \right) \left( \sum_{l=1}^{d} \mathbf{g}_j^i[r][l]\mathbf{w}_l \right) \right] = \mathbb{E}\left[ \sum_{l_1,l_2} \mathbf{g}_j^i[r][l_1]\mathbf{x}_{l_1} \mathbf{g}_j^i[r][l_2]\mathbf{w}_{l_2} \right] = $$
$$\sum_{l_1,l_2} \mathbb{E}[\mathbf{g}_j^i[r][l_1]\mathbf{g}_j^i[r][l_2]]\mathbf{x}_{l_1}\mathbf{w}_{l_2} = \sum_{l=1}^{d} \mathbb{E}[(\mathbf{g}_j^i[r][l])^2]\mathbf{x}_l\mathbf{w}_l = \mathbf{w}^\top \mathbf{x} \tag{14}$$

We used the fact that $\{\mathbf{g}_j^i[r][l]\}_{l=1,...,d}$ is a set of independent random variables for any fixed $i, j$ and $r = 1, ..., m$ and furthermore, each entry of $\mathbf{g}_j^i$ has zero mean and unit standard deviation. $\square$

We can thus conclude that:

$$\mathbb{E}[\text{EUGen}(\mathbf{w}, \mathbf{x})] = \sum_{i=0}^{k} a_i \frac{1}{m} \sum_{r=1}^{m} (\mathbf{w}^\top \mathbf{x})^i = \sum_{i=0}^{k} a_i (\mathbf{w}^\top \mathbf{x})^i = f(\mathbf{w}^\top \mathbf{x}) \tag{15}$$

That completes the proof. $\square$

## A.2 Proof of Theorem 3.2

*Proof.* We will leverage our analysis and notation from the proof of Theorem 3.1. We have:

$$\text{Var}(\text{EUGen}(\mathbf{w}, \mathbf{x})) = \frac{1}{m^2}\text{Var}\left(\sum_{i=0}^{k} T_i\right), \tag{16}$$

for $T_i$ defined as:

$$T_i = a_i \sum_{r=1}^{m} \prod_{j=1}^{i} \left(\sum_{l=1}^{d} \mathbf{g}_j^i[r][l]\mathbf{x}_l\right)\left(\sum_{l=1}^{d} \mathbf{g}_j^i[r][l]\mathbf{w}_l\right) \tag{17}$$

Note that by the conditions of the theorem, different $T_i$ are independent. Thus we can write:

$$\text{Var}(\text{EUGen}(\mathbf{w}, \mathbf{x})) = \frac{1}{m^2}\sum_{i=0}^{k}\text{Var}(T_i) \tag{18}$$

We can rewrite: $T_i = a_i \sum_{r=1}^{m} R_i$, where: $R_i = \prod_{j=1}^{i}\left(\sum_{l=1}^{d}\mathbf{g}_j^i[r][l]\mathbf{x}_l\right)\left(\sum_{l=1}^{d}\mathbf{g}_j^i[r][l]\mathbf{w}_l\right)$. Furthermore, again by the assumptions of the theorem, different $R_i$ are independent. Thus we have:

$$\text{Var}(T_i) = a_i^2 \sum_{r=1}^{m}\text{Var}(R_i) \tag{19}$$

Note that each $R_i$ has the same variance and thus we can write:

$$\text{Var}(\text{EUGen}(\mathbf{w}, \mathbf{x})) = \frac{1}{m}\sum_{i=0}^{k}a_i^2\text{Var}(R_1) \tag{20}$$

We have the following for $r = 1$:

$$\text{Var}(R_1) = \mathbb{E}[R_1^2] - (\mathbb{E}[R_1])^2 =$$
$$\mathbb{E}\left[\prod_{j=1}^{i}\left(\sum_{l=1}^{d}\mathbf{g}_j^i[r][l]\mathbf{x}_l\right)^2\left(\sum_{l=1}^{d}\mathbf{g}_j^i[r][l]\mathbf{w}_l\right)^2\right] - (\mathbf{w}^\top\mathbf{x})^{2i} = \tag{21}$$
$$\prod_{j=1}^{i}\mathbb{E}\left[\left(\sum_{l=1}^{d}\mathbf{g}_j^i[r][l]\mathbf{x}_l\right)^2\left(\sum_{l=1}^{d}\mathbf{g}_j^i[r][l]\mathbf{w}_l\right)^2\right] - (\mathbf{w}^\top\mathbf{x})^{2i}$$

It suffices to prove that:

$$\mathbb{E}\left[\left(\sum_{l=1}^{d}\mathbf{g}_j^i[r][l]\mathbf{x}_l\right)^2\left(\sum_{l=1}^{d}\mathbf{g}_j^i[r][l]\mathbf{w}_l\right)^2\right] = 2(\mathbf{w}^\top\mathbf{x})^2 + \|\mathbf{w}\|_2^2\|\mathbf{x}\|_2^2 + (\tau_{i,j} - 3)\sum_{l=1}^{d}\mathbf{w}_l^2\mathbf{x}_l^2 \tag{22}$$

That however follows by the computations completely analogous to those provided in the proof of Lemma 3.1 from [16]. □

## A.3 Proof of Theorem 3.3

*Proof.* Note that, by our previous analysis, we can rewrite:

$$\text{EUGen}(\mathbf{w}, \mathbf{x}) - f(\mathbf{w}^\top\mathbf{x}) = \sum_{i=0}^{k}\frac{1}{m}\sum_{r=1}^{m}Y_r^i, \tag{23}$$

where each $Y_r^i$ is defined as follows:

$$Y_r^i = a_i \prod_{j=1}^{i}\left(\sum_{l=1}^{d}\mathbf{g}_j^i[r][l]\mathbf{x}_l\right)\left(\sum_{l=1}^{d}\mathbf{g}_j^i[r][l]\mathbf{w}_l\right) - a_i(\mathbf{w}^\top\mathbf{x})^i \tag{24}$$

By our previous analysis, we know that $\mathbb{E}[Y_r^i] = 0$. Also, by the assumptions of the theorem, we can conclude that for any fixed $i$, $\{Y_r^i\}_{r=1}^m$ is an independent set of random variables. Denote: $K^i = \frac{1}{m} \sum_{r=1}^m Y_r^i$. Note that $K^0$ is completely deterministic. Therefore, by the union bound, we can write:

$$\mathbb{P}[|\text{EUGen}(\mathbf{w}, \mathbf{x}) - f(\mathbf{w}^\top \mathbf{x})| \geq \epsilon] \leq k \max_{i=1}^k \mathbb{P}[|K^i| \geq \frac{\epsilon}{k}] \tag{25}$$

By the assumptions of the theorem, the Cauchy-Schwarz and Triangle Inequality, we have the following:

$$|\frac{1}{m} Y_r^i| \leq \frac{1}{m} \left( |a_i| \prod_{j=1}^i (c\sqrt{d}\|\mathbf{x}\|_2)(c\sqrt{d}\|\mathbf{w}\|_2) + |a_i|(\mathbf{w}^\top \mathbf{x})^i \right) \tag{26}$$

Now we will apply the following version of the Azuma's Inequality:

**Lemma A.2** (Azuma's Inequality). *Assume that random variables $W_1, ..., W_m$ are independent and of mean zero. Assume furthermore that for some nonnegative $\{C_r\}_{r=1,...,m}$ we have: $|W_r| \leq C_r$ with probability one. Then the following holds:*

$$\mathbb{P}[|W_1 + ... + W_m| \geq \epsilon] \leq 2\exp(-\frac{\epsilon^2}{2\sum_{r=1}^m C_r^2}) \tag{27}$$

To complete the proof of Theorem 3.3, it suffices to apply the above Azuma's Inequality for random variable $W_i$ defined as: $W_r = \frac{1}{m} Y_r^i$ and Inequality 26. $\qquad \square$

### A.4  Proof of Theorem 3.4

*Proof.* We will apply the following well-known result, showing how Bernstein polynomials can be used to approximate continuous functions:

**Lemma A.3** (Bernstein Polynomials for Continuous Functions Approximation). *Take some constant $A > 0$. Let $f : [-A, A] \to \mathbb{R}$ be bounded. Then there exists $C > 0$ such that the following holds:*

$$\|f - B_n(f)\|_\infty \leq C\omega(\frac{1}{\sqrt{n}}), \tag{28}$$

*where $B_n(f)$ is the nth Bernstein polynomial of $f$ (in particular of degree $n$).*

To complete the proof of Theorem 3.4, it suffices to: (1) note that $p_i$ is the upper bound of $\mathbb{P}[|\text{EUGen}(\mathbf{w}, \mathbf{x}) - f(\mathbf{w}^\top \mathbf{x})| \geq \epsilon]$ (this is implied by Theorem 3.2 and Chebyshev's Inequality), apply Theorem 3.3 and Triangle Inequality. $\qquad \square$

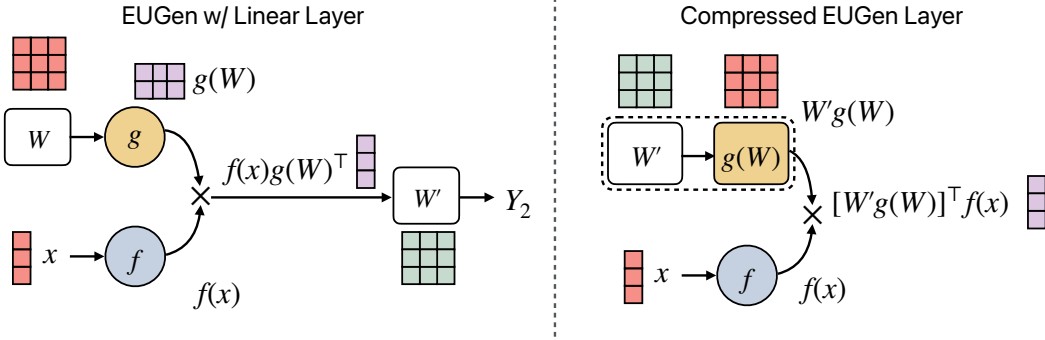

Figure 11: Schematic diagram of our compression method. (Top row) : A EUGen layer followed by a linear layer. (Bottom Row) : The weights can be multiplied together to create a single weight matrix without affecting the inputs, effectively disentangling the weights and the inputs.

# B    Extended Functionalities of EUGens

In this section, we discuss additional properties of the EUGen layers and show how certain special instantiations generalize several well-known layers and methods.

## B.1    Compressing Networks by Combining Layers

For training NeRF models and uptraining Transformers, we use an EUGen layer followed by a linear layer. In this section, we explain how we can achieve improved efficiency by compressing linear layers with EUGens. Recall our EUGen layer is given by the following equation (following the PyTorch convention) :

$$\mathbf{Y}_1 = \Phi(\mathbf{X})\Psi(\mathbf{W})^\top \tag{29}$$

If the EUGen layer is followed by the linear layer with weight $\mathbf{W}'$ and bias $\mathbf{b}'$, then the results is:

$$\begin{aligned}
\mathbf{Y}_2 &= \mathbf{Y_1}\mathbf{W}'^\top + \mathbf{b}' \\
&= \Phi(\mathbf{X})\Psi(\mathbf{W})^\top\mathbf{W}'^\top + \mathbf{b}' \\
&= \Phi(\mathbf{X_1})\hat{\mathbf{W}}^\top + \mathbf{b}',
\end{aligned}$$

where $\hat{\mathbf{W}} = \mathbf{W}'\Psi(\mathbf{W})$. This simple compression technique allows us to collapse any EUGen layer followed by a FFL in our experiments, allowing for more efficient inference (see Fig. 11).

## B.2    Closed-Form Knowledge Distillation

The formulation of the EUGen layer makes knowledge distillation particularly simple. Specifically, we want to train an EUGen layer $\text{EUGen}^k(\mathbf{W}, \cdot)$ to match the outputs of a trained FFL using MSE as the metric. For brevity, we will suppress $k$ for the rest of the subsection. If $\mathbf{x}$ (resp. $\mathbf{y}$) is the input (resp. output of a trained FFL, we want to find the weights $\hat{\mathbf{W}}$, which minimize the following error :

$$\min_{\mathbf{W}} \|\text{EUGen}(\mathbf{W}, \mathbf{x}) - \mathbf{y}\|_2 \tag{30}$$

The optimal $\hat{\mathbf{W}}$ has a closed form :

$$\hat{\mathbf{W}} = (\mathbf{x}'^\top\mathbf{x}')^{-1}\mathbf{x}'^\top\mathbf{y} \tag{31}$$

where $\mathbf{x}' := \Phi(\mathbf{x})$.

The above solution is unique iff $\mathbf{x}'^\top\mathbf{x}'$ is invertible. In the case where $\mathbf{x}'^\top\mathbf{x}'$ is not invertible, the inverse can be replaced by the Moore-Penrose pseudoinverse. Note that, there also exists a closed form formula for regression with weight decay ($l_2$-regularization), i.e. solution for the following optimization problem :

$$\|\text{EUGen}(\mathbf{W}, \mathbf{x}) - \mathbf{y}\|_2 + \lambda\|\mathbf{W}\|_2$$

Thus, the problem of distilling specific input-output pairs in the EUGen layer can be solved *without backpropagation*. Note that an analytic formula does not exist if the projection matrices $\mathbf{G}$ are trainable. However, in that case, the optimization is extremely lightweight and can be performed with minimal compute.

## B.3 Connections with Low-Rank Layers and 2-layer Neural Networks

In this section, we will discuss connections with our layer with low rank matrices, asymmetric kernels, SNNKs and works on 2-layer Neural Networks. Finally we show some results on the expressivity of some special instantiations of EUGen layers.

Recall that our EUGen layer is of the form $\text{EUGen}^k(\mathbf{W}, \mathbf{x}) = \Psi_{\mathbf{G}}(\mathbf{W})\Phi_{\mathbf{G}}(\mathbf{x})$, where $\mathbf{G}$ is a (Gaussian) projection matrix. In practice, we learn both $\mathbf{W}$ and $\mathbf{G}$, making the EUGen mechanism quite expressive. This setup naturally leads to a more general variant: $\text{EUGen}^k_+(\mathbf{W}, \mathbf{x}) = \Psi_{\mathbf{H}}(\mathbf{W})\Phi_{\mathbf{G}}(\mathbf{x})$ where $\mathbf{H}$ and $\mathbf{G}$ are different project matrices. While $\mathbf{H}$ may not be explicitly parameterized, its effect can be captured implicitly by learning $\mathbf{W}$, since EUGen involves computing products like $\mathbf{GW}$. Empirically, we find in some cases that using separate projection matrices for transforming $\mathbf{W}$ and $\mathbf{x}$ in the EUGen layer may be beneficial during training. Thus, we can consider the explicit variant: $\text{EUGen}^k(\mathbf{W}, \mathbf{x}) = \Psi_{\mathbf{H}}(\mathbf{W})\Phi_{\mathbf{G}}(\mathbf{x})$, which allows for a more flexible form of *kernel learning* through disentangled projections. A similar idea is explored in [122].

With this formulation, we are now ready to show how EUGen generalizes various well-known approaches.

We start by showing how low rank layers can be a special case of our EUGen layers. In this case, take $\Psi = \Phi = Id$, $\mathbf{H} = \mathbb{I}$ and $k = 1$, and in this case $\text{EUGen}(\mathbf{W}, \mathbf{x}) = \mathbf{WGx}$.

Then $\mathbf{G}$ is low rank, EuGen degenerates into a low rank layer. Note that these layers have been used in the training of neural networks [70, 108, 81]. Similar to observations in Wei et al. [108], we find that the low rank layers struggle to yield good performance, and using a non-linear $\Phi$ leads to quality gains (almost $9\%$), even with a $\Psi$ as the identity mapping.

When $\Phi$ and $\Psi$ are related to the Fourier transform of the activation function of the feed-forward layer (see [80] for precise definitions of $\Phi$ and $\Psi$) and $\mathbf{G} = \mathbf{H}$, then EuGen becomes the SNNK layer. However, SNNKs cannot unbiasedly approximate FFLs with unbounded activations, which is not the case for EuGeNs (Theorem 3.1). EuGeNs can accurately and unbiasedly approximate polynomial-activation FFLs without any learning, simply by sampling matrices $G$ from predefined distributions. During training, SNNKs may implicitly approximate various FFLs; however, such approximations remain biased.

Next, we explain why we view our EUGen layer as an asymmetric kernel between the space of weights and the space of inputs. We interpret the EUGen layer as computing a similarity measure between the weights and the inputs, after projecting them into a lower-dimensional space. The motivation for the lower dimensionality of the weights comes from Mousavi-Hosseini et al. [62] and this can lead to better generalizability.

For convenience of the reader, following Wu et al. [109], we define asymmetric kernels.

**Definition B.1.** (Asymmetric Kernel). Let $\mathcal{X}$ and $\mathcal{Y}$ be two input spaces, and $\mathcal{H}$ be feature space assumed to be a Hilbert space. Asymmetric kernel is a function $k : \mathcal{X} \times \mathcal{Y} \to \mathbb{R}$, satisfying $k(x, y) = \langle \phi_{\mathcal{X}}(x), \phi_{\mathcal{Y}}(y) \rangle_{\mathcal{H}}$ for all $x \in \mathcal{X}$ and $y \in \mathcal{Y}$, where $\phi_{\mathcal{X}}$ and $\phi_{\mathcal{Y}}$ are mapping functions from $\mathcal{X}$ and $\mathcal{Y}$ to $\mathcal{H}$, respectively.

From the definition, it is clear that our EUGen-layers are indeed asymmetric kernels between the space of weights and the space of inputs.

As explained earlier, even a simple $\Psi$, for example, the identity function can lead to fairly expressive networks which we will now show below by incorporating deep and powerful results on 2-layer ReLU networks. For simplicity of the discussion, let us assume our EUGen layer is of the form : $\mathbf{EUGen}(\mathbf{W}, \mathbf{x}) = \sum_{i=1}^r w_i f(\langle \mathbf{g_i}, \mathbf{x} \rangle), w_i \in \mathbb{R}, g_i \in \mathbb{R}^d$, and $f$ is the ReLU activation, i.e. $\Psi$ is the identity function, $\mathbf{H} = \mathbb{I}$ and $\Phi(\mathbf{x}) := \text{ReLU}(\mathbf{Gx})$.

In this special case, EUGen corresponds to the first-order Taylor expansion of 2-layer neural network with respect to the top-layer weights $\mathbf{W}$.

The following the results can be easily extended to $\mathbb{R}^k$.

**Proposition B.2.** *(1) Under certain conditions, EUGen layers without trainable* $\mathbf{G}$ *and* $r$ *number of random features can approximate any polynomial of degree at most* $l$, *where* $l^{1+\delta} < r \leq l^{2+\delta}$ *for some* $\delta > 0$.
*(2) Every* 1-*Lipschitz function can be approximated with respect to the* $l^2$ *norm over* $[-1,1]^d$ *by a EUGen-layer of poly($r$) number of random features.*

*Proof.* (1) is proved in Theorem 1 in Ghorbani et al. [34], while (2) can be shown via Theorem 1 and 2 in Hsu et al. [41]. □

When $\mathbf{G}$ is made trainable, even simple EUGen-layers effectively function as 2-layer ReLU networks, inheriting their strong expressive power (for ex: See Theorem 1 in Boursier et al. [6]). Moreover, it is shown that the 2-layer networks are more expressive than their degree 1 Taylor series counterparts [34].

Finally we would like to clarify the differences between our work and the works on the linearization of 2-layer networks.

A lot of work has been focused on the linearization of 2-layer networks of the form $\mathbf{L}(\theta, \mathbf{x}) = \mathbf{W_2} f(\mathbf{W_1 x})$, where $\theta = (\mathbf{W_1}, \mathbf{W_2})$, and $f$ is some non-linear function and $\mathbf{W_1}$ is fixed and *only* $\mathbf{W_2}$ is trainable. Under various assumptions, one can express $\mathbf{L}$ as a kernel [14, 13, 34, 7, 65]. Note that our goal is fundamentally different as we are trying to linearize a *single* layer and not a 2-layer network of a specific form. Moreover, allowing the projection matrices $\mathbf{G}$ to be trainable, our EUGen layers do not fit in these above frameworks.

## C  Additional Related Works on Implicit Neural Representations

In this section, we discuss techniques commonly used to accelerate NeRFs and SDFs. Neural Radiance Fields (NeRF) [61] achieve remarkable quality in 3D reconstruction by leveraging MLPs to map spatial points to volume densities and colors. However, both training and rendering processes remain time-consuming, prompting extensive research aimed at accelerating these tasks, by integrating explicit representations [55, 118, 42, 89, 87, 11, 84, 64, 31, 112, 5, 88, 43], dividing a scene into smaller blocks [75, 89, 95, 110, 96], caching (baking) the implicit functions [32, 39, 12, 76], and devising some novel tweaks [54, 37].

Signed Distance Fields (SDF) are scalar fields that represent the distance to the nearest surface point, crucial in robotics for tasks such as environment mapping, collision avoidance, and trajectory optimization [127, 93, 79, 63, 28, 27, 10, 20, 98, 123, 45, 24, 113, 126]. In computer vision, Truncated Signed Distance Fields (TSDF) are employed in SLAM systems, as traditional SDF computations can be prohibitively expensive in real time [66, 67]. Several approaches have emerged to facilitate real-time SDF construction due to its importance in robotics [69, 38, 68], while others have addressed challenges in constructing SDF for complex environments [30, 74, 33, 72]. To get a photo-realistic quality of scene reconstruction, recent works replace the volume density in NeRF formulations with SDF, using well-designed SDF-to-volume density conversion formulas [114, 104, 115]. Further acceleration is achieved by combining explicit grid structures with implicit SDF [120, 105, 52, 103, 77], populating thin shells near the zero level set [106], or baking SDF into a high-quality triangle mesh [116].

# D  Experiments

**Contents**

In this section, we provide additional details for the experiments in our main paper. Our code for the EUGen layer implementation and related experiments can be found at `https://github.com/arijitthegame/EUGen/`.

## D.1  Synthetic experiments.

For these experiments, 10,000 samples are drawn randomly from the range $(0, 1)$ with a dimension of 512, and are then passed through a Linear layer of shape $(512, 512)$. Non-linearity of ReLU (resp. Softplus) is applied on the outputs of the linear layer. Our aim is to show that our EUGen-layer can accurately approximate these outputs.

We use the SNNK layers [80] and Low Rank approximation as baselines. We train EUGen and SNNK layers for 3k epochs with 128 random features using the Adam optimizer [48]. We repeat this experiment 10 times and report the MSE loss across the replicates. This experiment is run on a single NVIDIA RTX 4090.

We present an additional experiment on image fitting by replacing FFLs in the Siren network [82] which is a MLP with sine activations. The Siren network for the image fitting experiment is a 3 layer neural network with hidden dimension matrices of sizes $256 \times 256$. It was shown in the original work that Siren network can not only fit the image but can also accurately represent the derivatives of the signal. We replace the hidden layer with our EUGen layer with 32 random features, resulting in modest computational gains. We show that like the original Siren network, our EUGen-

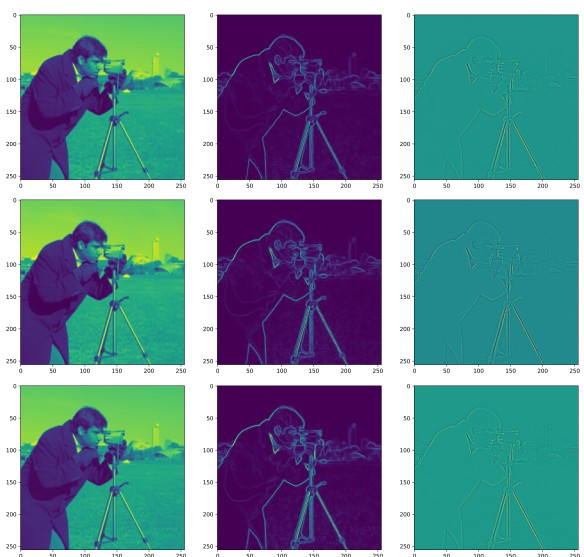

Figure 12: Top to bottom : The baseline Siren network on the first row, fitting not only the image, but also the gradients. (Middle) : SNNK-Siren (Bottom) EuGen-Siren on the bottom row produces an accurate approximation of the above.

Siren network can also accurately model the derivatives of the signal. It should be noted that our EUGen-Siren network is about $\frac{1}{3}$ the size of the baseline Siren network.

## D.2  Detailed results for 3D scene reconstructions

In this subsection, we provide details for all 3D reconstruction experiments in the main paper.

**NeRF Experiments.**  For these experiments, we use the default configuration of the Pytorch implementation of NeRF [117]. The inference time is computed using a single NVIDIA RTX 4090 and an AMD Ryzen 7 7700 8-Core Processor. All tests are conducted on the Synthetic NeRF dataset [61], and the details of our architecture choices are illustrated in Figure 13 (top row). In our

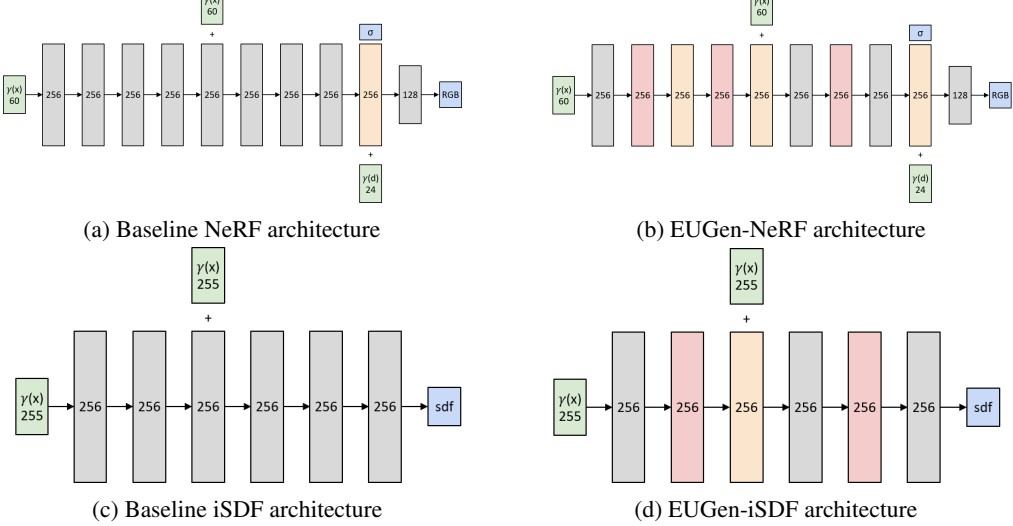

(a) Baseline NeRF architecture        (b) EUGen-NeRF architecture

(c) Baseline iSDF architecture        (d) EUGen-iSDF architecture

Figure 13: **(Top)** : Network architecture of the baseline NeRF and our EUGen-NeRF model. **Gray** denotes Linear ReLU layer, **Orange** is Linear without activation, **Pink** is the EUGen layer, + is concatenation, and the activation for the RGB output is Softmax. **(Bottom)** Network architecture of the SDF prediction network in iSDF. (b) Our modified EUGen-iSDF architecture. Note that in both cases, EUGen layer followed by the Linear layer can be compressed for more efficient network inference.

experiments, we set the number of random features to 256, with additional results for the different number of random features provided in Section E.2.

We report the detailed results for each scene in Table 1. We achieve an almost **24**% increase in inference speed and lower storage costs, while matching the performance of the baseline.

**iSDF Experiments.** Figure 13 (bottom row) shows the baseline iSDF [69] and the modified EUGen-iSDF architectures. We use six ReplicaCAD [86] scenes, following original iSDF [69] paper. In all experiments, we use the default configuration of public available iSDF code while replacing only the SDF network with our EUGen-iSDF.

To evaluate efficiency, we report two time-related metrics: training time and inference time. However, it's important to note that, as iSDF is an incremental mapping module, the number of keyframes can vary between different experiment runs. Because the sample size is proportional to the number of keyframes, training time may not serve as a perfect measure of the network's efficiency. To account for this, we also report inference time, which is measured by sampling 1000 random rays and recording the time taken for a single forward pass through the network.

In Figure 21, we visualize the reconstructed SDFs for all 6 ReplicaCAD dataset and report the quantitative results in Table 4. We achieve **23**% increase in inference speed while matching the quality of the baseline.

**Mip-NeRF 360 Experiments.** Mip-NeRF 360 [4] is one of SOTA models for novel view synthesis. We use the NeRF-Factory PyTorch implementation of Mip-NeRF 360. To fit within memory constraints, we reduce the training batch-size from 4096 to 2048. Experiments are conducted on the *360_v2* dataset, replacing three network layers with EUGen layers, each using 1024 random features.

Inference is performed on a single NVIDIA RTX 4090 GPU with an AMD Ryzen 7 7700 CPU.

We report the detailed results for each scene in Table 2. EUGen layers produce comparable results while being **27**% faster.

**Zip-NeRF Experiments.** We build on prior work that accelerates NeRF by combining explicit hash-grid structures with MLPs [11, 84, 64, 5], enabling more efficient spatial encoding and reducing MLP overhead. However, these improvements do not address inefficiencies within the MLPs themselves. We tackle this gap using our EUGen layers.

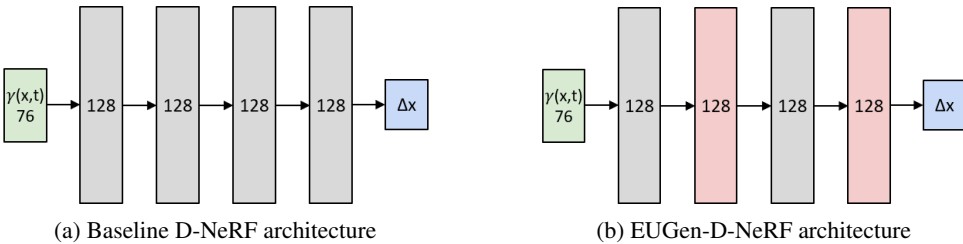

(a) Baseline D-NeRF architecture          (b) EUGen-D-NeRF architecture

Figure 14: (a) Architecture of the deformation network in D-NeRF [71, 90]. (b) Our modified EUGen-D-NeRF architecture. **Gray** blocks denote Linear ReLU layers, Pink blocks are the EUGen layers, and there is no nonlinear activation (ReLU) for $\Delta x$.

We focus on Zip-NeRF [5], a state-of-the-art method for unbounded scenes that uses shallow MLPs to decode grid-based features. Its architecture contains two proposal MLPs for sample placement and a NeRF MLP for producing color outputs. Due to the proposal MLPs' low dimensionality ($6 \rightarrow 64 \rightarrow 1$), we instead target the NeRF MLP, specifically its view-dependent component composed of two linear ReLU layers. We replace the first of these with our EUGen layer, resulting in EUGen-Zip-NeRF—a minimal intervention that serves as a strong case study for EUGen's effectiveness.

All experiments use the default PyTorch implementation of Zip-NeRF [36] on the Mip-NeRF 360 dataset [4]. To match NeRF rendering benchmarks, we use consistent hardware settings but reduce the ray batch size from 65,536 to 4,096 due to memory constraints. We use 64 random features for the EUGen layer.

All rendered images from Mip-NeRF 360 dataset can be shown in Figure 16. Despite this minimal change, we maintain similar rendering quality while improving the rendering speed by **6**%.

**Dynamic NeRF Experiments.**   In this experiment, we evaluate our method on D-NeRF [71], which synthesizes novel views of dynamic scenes with complex, non-rigid motion. D-NeRF extends NeRF by introducing a deformation network that takes position and time as inputs and outputs a 3D offset, effectively bending rays to model scene motion over time.

We use the PyTorch implementation of D-NeRF [90], which incorporates a hash-grid structure [64] to accelerate training and rendering. As a result, the network is shallower than the original D-NeRF. To apply our method, we replace two layers in the deformation network with EUGen layers (see Figure 14).

For a fair comparison with the baseline, we reduce randomness by using the `preload` configuration and disabling other acceleration options, ensuring consistent settings across all experiments.

We present the rendered images in Fig. 19. We achieve a **7**% speedup in inference while maintaining a similar reconstruction quality.

Table 1: Comparison between NeRF and our EUGen-NeRF on the Synthetic NeRF dataset [61]. We report PSNR, SSIM, and LPIPS per scene, and also include the averaged results along with inference time and storage. A higher PSNR/SSIM and a lower LPIPS/Time/Storage is preferred.

|  |  | Chair | Drums | Ficus | Hotdog | Lego | Materials | Mic | Ship | Avg. | Time ↓ | Storage ↓ |
|---|---|---|---|---|---|---|---|---|---|---|---|---|
| PSNR ↑ | NeRF | 33.70 | 25.46 | 26.40 | 35.81 | 31.31 | 29.00 | 33.11 | 28.79 | **30.45** | 2.36s | 3.27MB |
|  | EUGen-NeRF | 33.09 | 25.18 | 28.40 | 35.47 | 30.60 | 28.88 | 32.63 | 28.66 | 30.36 | **1.79**s | **2.27**MB |
|  | w/o trainable **G** | 32.76 | 24.92 | 28.11 | 35.22 | 30.14 | 28.62 | 32.42 | 28.61 | 30.10 | **1.79**s | **2.27**MB |
| SSIM ↑ | NeRF | 0.978 | 0.930 | 0.939 | 0.979 | 0.965 | 0.956 | 0.979 | 0.870 | **0.950** | - | - |
|  | EUGen-NeRF | 0.974 | 0.925 | 0.961 | 0.977 | 0.957 | 0.954 | 0.976 | 0.863 | 0.949 | - | - |
|  | w/o trainable **G** | 0.972 | 0.922 | 0.958 | 0.975 | 0.952 | 0.951 | 0.975 | 0.857 | 0.945 | - | - |
| LPIPS ↓ | NeRF | 0.013 | 0.049 | 0.048 | 0.012 | 0.017 | 0.021 | 0.020 | 0.074 | **0.032** | - | - |
|  | EUGen-NeRF | 0.016 | 0.053 | 0.023 | 0.015 | 0.022 | 0.023 | 0.022 | 0.079 | **0.032** | - | - |
|  | w/o trainable **G** | 0.019 | 0.056 | 0.025 | 0.017 | 0.025 | 0.025 | 0.026 | 0.084 | 0.035 | - | - |

**Distillation Results.**   We show complete distillation results for NeRF in Table 5 on the *lego* dataset.

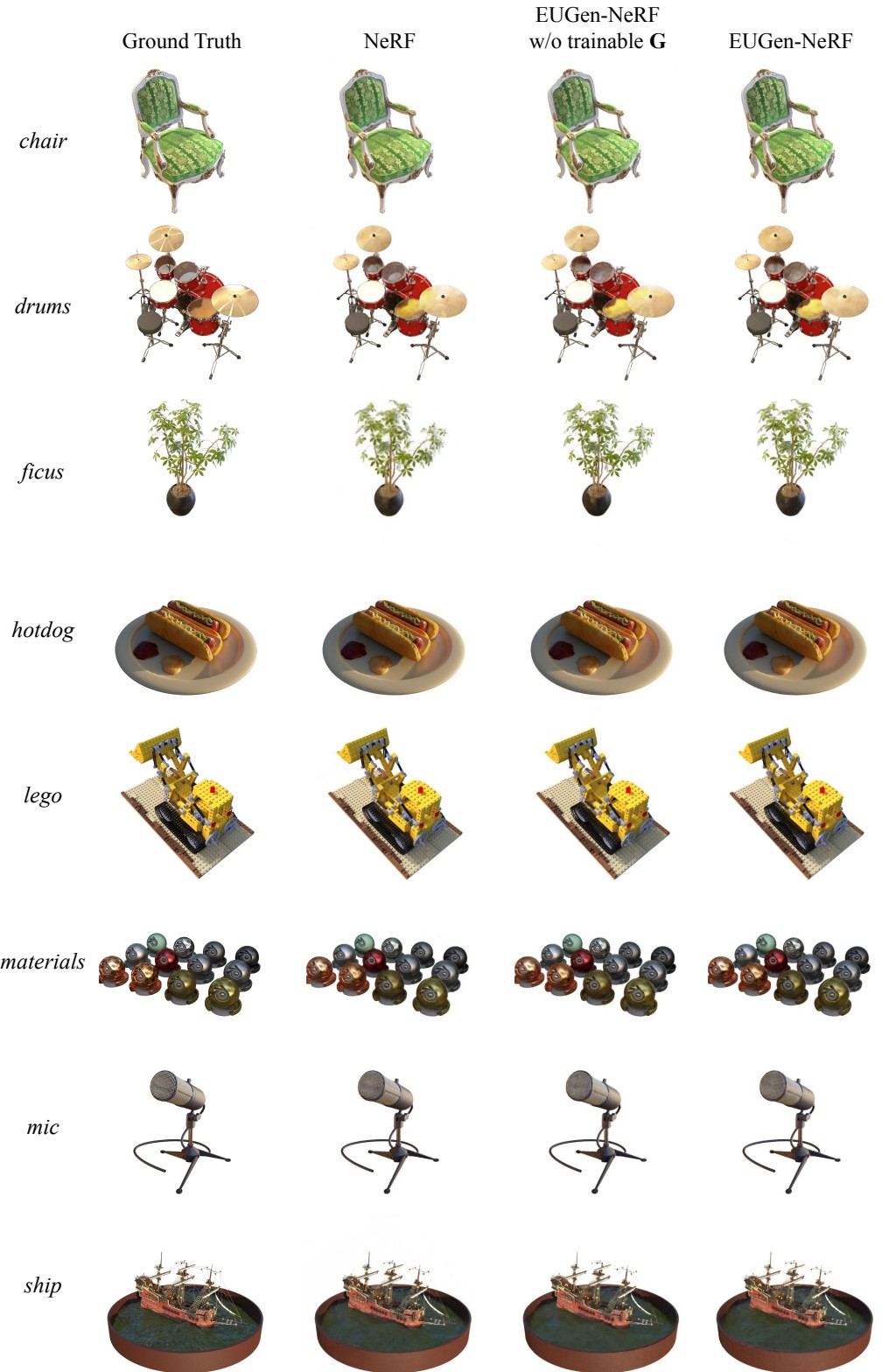

Figure 15: Rendered images from the Synthetic NeRF dataset [61]. For each scene, the images from left to right show the Ground Truth, NeRF [61], and EUGen-NeRF.

Table 2: PSNR of rendered images for Mip-NeRF 360 in *360* V2 dataset. In all settings, a higher PSNR & SSIM, and a lower LPIPS is preferred. We observe that EUGen achieves comparable performance to Mip-NeRF while being **27**% faster.

|  | Scene | Bicycle | Bonsai | Counter | Garden | Kitchen | Room | Stump |
|---|---|---|---|---|---|---|---|---|
| PSNR (↑) | Mip-NeRF 360 | 22.31 | 30.59 | 27.77 | 24.65 | 29.87 | 31.18 | 24.49 |
|  | EUGen-Mip-NeRF 360 | 22.10 | 29.99 | 27.29 | 24.30 | 29.34 | 30.60 | 24.04 |
| SSIM (↑) | Mip-NeRF 360 | 0.466 | 0.897 | 0.809 | 0.626 | 0.879 | 0.885 | 0.569 |
|  | EUGen-Mip-NeRF 360 | 0.435 | 0.882 | 0.793 | 0.593 | 0.859 | 0.869 | 0.538 |
| LPIPS (↓) | Mip-NeRF 360 | 0.529 | 0.230 | 0.301 | 0.364 | 0.169 | 0.262 | 0.488 |
|  | EUGen-Mip-NeRF 360 | 0.557 | 0.263 | 0.328 | 0.389 | 0.196 | 0.295 | 0.516 |

Table 3: Detailed hyperparameters for EUGen-ViT experiments on ImageNet and Places365 datasets. LR stands for learning rate.

| Hyperparameter | **ImageNet** (1.28M) | **Places365** (1.8M) |
|---|---|---|
| Num. layers | 12 | 12 |
| Num. heads | 12 | 12 |
| Hidden size | 768 | 768 |
| MLP dim. | 3072 | 3072 |
| Num. patches | $16 \times 16$ | $16 \times 16$ |
| Batch size | 4096 | 4096 |
| Epochs / Total # Steps | ef7300 epochs | $80,000$ steps |
| Base LR | $10^{-3}$ | $10^{-3}$ |
| Final LR | $10^{-5}$ | $10^{-5}$ |
| Optimizer | Adam | Adam |
| LR schedule | Linear warmup ($10^4$ steps - Imagenet), 500 steps - Places365), constant, cosine decay | |

In Figure 20 (left), we visualize the rendered image with the distilled Zip-NeRF model. We set the number of random features to 64, with ablations provided in Appendix E.2 (Fig. 27 and Table 10). Furthermore, to make the distillation more efficient, we adopt several strategies: (1) we sample only 1/16 of the training images, (2) we render them at 1/8 of the original resolution, and (3) we randomly sub-sample 10% of the resulting input-output pairs. These choices significantly reduce the memory footprint and allow the distillation process to run efficiently within our hardware constraints.

We also test the distillation in the last hidden layer of iSDF [69]. In Fig. 20 (right), we visualize the reconstructed SDF and the extracted zero level-set mesh from *apt_2_nav* dataset. Our results indicate that the reconstruction quality of the distilled EUGen-iSDF is comparable to that of the baseline iSDF, with a **9**% improvement in inference speed.

### D.3 Image Classification Experiments

In this section we list the experiment setup of all image classification experiments. First we provide the hyperparameters and experiment setup for ImageNet and Places365. We replace a part of the Feed-Forward Network (FFN) block in Transformers with the EUGen layer, namely the expansion layer with the GeLU activation. Note that the EUGen layer can then be combined with the following linear layer to create a single linear layer of size $[r, 768]$, where $r$ is the number of random features. This allows for a significant compression of the ViT model.

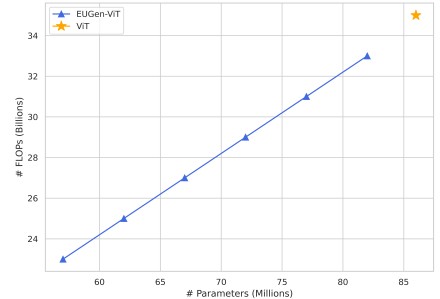

Figure 18: Flops vs parameters. EUG-ViT obtains a **30**% reduction in flops.

The hyperparameters and model architecture are shown in Tab. 3. Fig. 4 illustrates that we achieve approximately the same accuracy as ViT full (86M parameters) by replacing its top-6 MLP layers with EUGen, leading to a **30**% reduction in inference parameters.

Table 4: Quantitative results of EUGen-iSDF. RF is the number of random features, L1$_{avg}$ is calculated across all SDF levels, and L1$_{surf}$ is for ground truth surface points. $t_{train}$ refers to iteration time (including backward pass), and $t_{infer}$ measures forward pass inference time. Results are averaged across 6 ReplicaCAD [86] sequences with 3 random seeds. Our EUGen-iSDF model achieves less than similar quality as the baseline while being **23**% faster.

| | Distance (cm) | | | Time (ms) | |
|---|---|---|---|---|---|
| Method | L1$_{avg}$ ↓ | L1$_{surf}$ ↓ | CD ↓ | $t_{train}$ ↓ | $t_{infer}$ ↓ |
| iSDF | 8.99 | 4.08 | 2.02 | 8.24 | 1.37 |
| EUGen-iSDF | 9.37 | 4.17 | 2.10 | 7.81 | 1.06 |

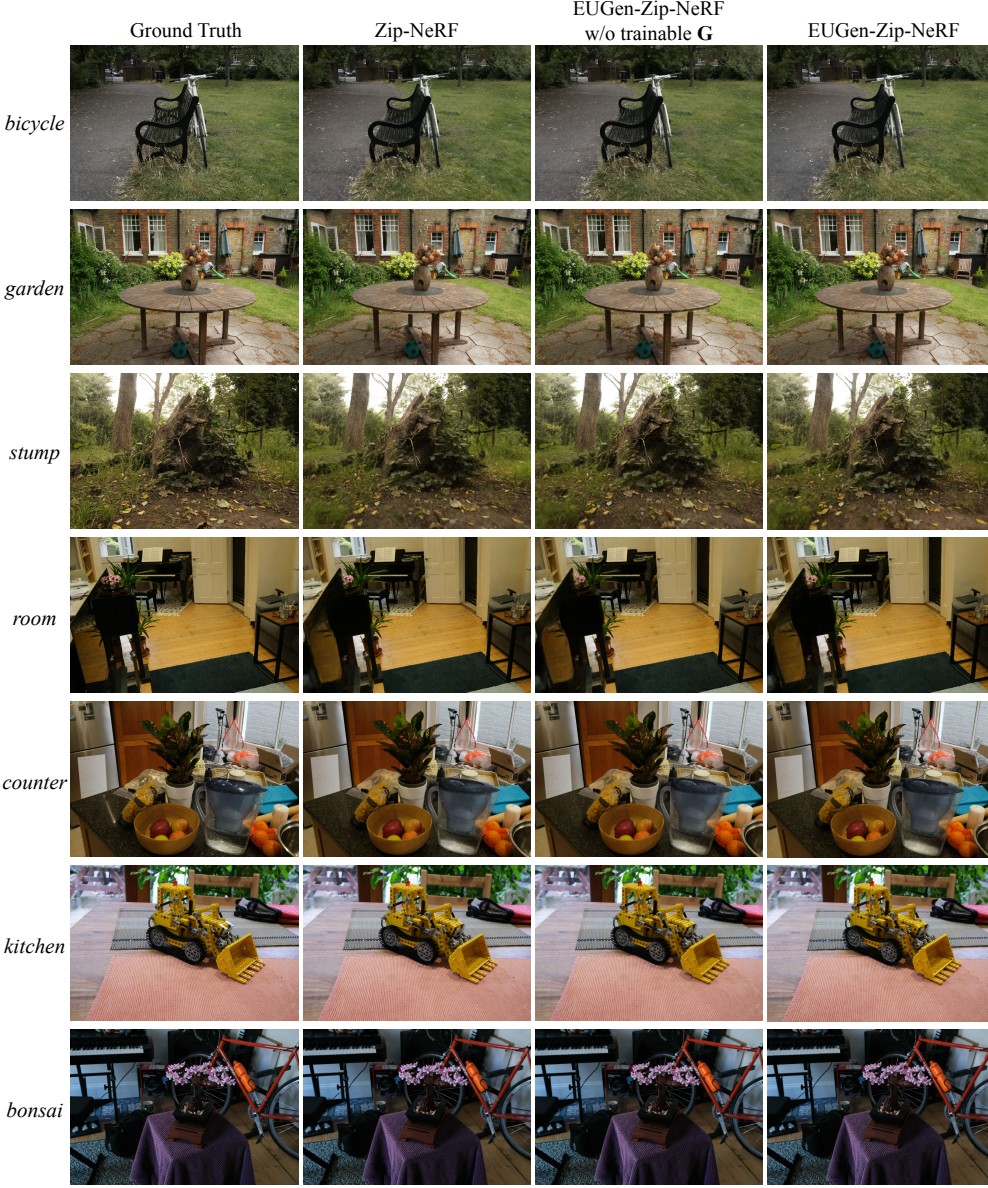

Figure 16: Rendered images from the Mip-NeRF 360 dataset [4]. For each scene, the images from left to right show the Ground Truth, Zip-NeRF [5], and EUGen-Zip-NeRF.

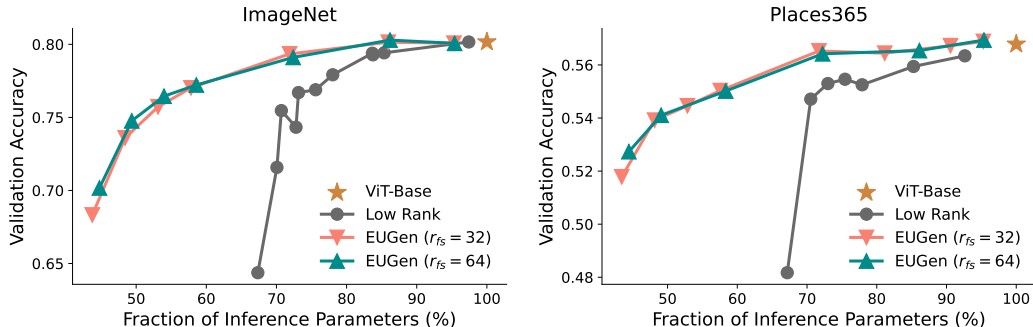

Figure 17: Evaluation results of EUGen for image classification tasks: (*left*) ImageNet and (*right*) Places365 datasets using $ViT_{base}$ (86M parameters). We observe that EUGens can match the performance of vanilla ViTs with a significantly smaller fraction of parameters than the Low-Rank baseline.

Table 5: Distillation results for NeRF in *lego* dataset. Trainable **G** produces a leaner efficient model while being almost **26**% faster. Both analytic and trainable uses 128 random features except the last row which uses 64 random features.

| Method | PSNR ↑ | SSIM ↑ | LPIPS ↓ | Inference Speed (s) |
|---|---|---|---|---|
| Baseline | 31.31 | 0.965 | 0.017 | 2.36 |
| Analytic | 28.54 | 0.948 | 0.027 | 1.84 |
| Trainable **G** | 31.25 | 0.965 | 0.017 | 1.84 |
| Trainable **G** (64) | 31.09 | 0.964 | 0.017 | 1.75 |

Moreover in Fig. 17, we show that we can improve accuracy by increasing the number of random features and our models greatly outperform the low rank variants.

Next we show results on additional image classification tasks, namely on Cifar-10, Cifar-100, DTD and Oxford Pets. For these set of experiments $r = 16$. Fig. 22 shows that we can maintain accuracy across a wide range of image classification tasks using ViT while reducing the size of the model by almost **40**% and improving inference speeds by **34**%. We present a detailed analysis of flops in Fig. 18 as we successively replace MLP layers starting from the top layer.

To summarize, by replacing top-6 layer's MLP blocks with EUGen reduces the size of the ViT model from 346 Mb to **196.85** Mb. This speeds up inference by almost **35**% and has minimal impact on accuracy. We use the AdamW [58] optimizer with a weight decay of $10^{-5}$ and a learning rate of $3 \times 10^{-5}$ and a constant scheduler with warmup steps $6$% of the total number of training steps with a batch size of 16.

Moreover our method can be combined with other efficient architectures like EfficientViT [56], showcasing the versatility of our method.

**Scaling EUGens to Deeper Models :** In this subsection, we will show that EUGens can be used in deeper models like ViT-L which has 24 layers. Table 6 shows that replacing most ViT-L MLP blocks with EUGens yields over a **50**% reduction in inference parameters, matching performance on Places365 and incurring only a small drop on ImageNet.

Table 6: Top-1 accuracy (%) and parameter counts for baseline and EUGen variants. EUGen-$x$ corresponds to $x$ MLP blocks in Vit-L replaced by EUGen.

| Dataset | Baseline | EUGen-18 | EUGen-21 | Params (B / 18 / 21) |
|---|---|---|---|---|
| Places365 | 58.60 | 57.47 | 57.04 | 304M / 176M / 150M |
| ImageNet | 82.20 | 78.91 | 79.13 | 304M / 176M / 150M |

Table 7: EuGen experiments on GLUE benchmarks on Linearizing the Tanh Pooler Layer. MCC score is reported for CoLA, F1 score is reported for MRPC and QQP, Spearman correlation is reported for STSB, and accuracy scores are reported for the other tasks. All results are an average of 5 seeds. Our EUGen model beats all the other baselines on all the GLUE tasks.

| Dataset | # Training Parameters | RTE | MRPC | QNLI | QQP | SST-2 | MNLI | STSB | COLA |
|---|---|---|---|---|---|---|---|---|---|
| Pooler + Classifier | $\sim 59k$ | 57.5 | 81.5 | 74.5 | 72.0 | 84.9 | 56.4 | 78.1 | 29.4 |
| Linear Probe | $\sim 1k$ | 57.8 | 81.2 | 69.2 | 62.2 | 83.1 | 43.4 | 69.9 | 35.4 |
| SNNK [80] | $\sim 2k$ | 60.1 | 81.5 | 72.2 | 70.2 | 83.0 | 53.7 | 71.8 | 32.8 |
| EuGen (ours) | $\sim 2k$ | **61.7** | **83.8** | **75.5** | **72.8** | **86.8** | **57.4** | **82.8** | **45.2** |

## D.4  NLP Experiments

In this section, we provide additional NLP experiments, namely uptraining BERT [25] on the GLUE benchmark [102].

**Linearizing the Feed-Forward Networks**     Similar to the GPT setting, we replace a part of the Feed-Forward Network (FFN) block in Transformers with the EUGen layer, namely the expansion layer with the GeLU activation. Our compression trick allows for a significant reduction of parameters of the BERT model. In all our experiments $r = 32$. We report the MCC score for CoLA, F1 score for MRPC and QQP, Spearman correlation for STSB, and accuracy scores for the other tasks.in Fig. 22 as we successively replace MLP layers starting from the top layer. We use the AdamW optimizer [58] with a weight decay of $10^{-5}$ and a learning rate of $3 \times 10^{-5}$, and a constant scheduler with warmup steps $6\%$ of the total number of training steps with a batch size of 16.

We also present the detailed results for each GLUE task in Figure 24. Our EUGen-BERT model matches the baseline model across all the 8 tasks while being almost **26**% smaller. We also show that EUGens can accelerate smaller language models like DistilBERT. Our results show that we can further compress DistilBERT by **21**%. Thus our method can complement other well-known methods for model compression like knowledge distillation and pruning.

**Linearizing the Pooler Layer**    Many encoder only pretrained transformers like BERT use a special token called [CLS] that is appended in front of the input sequence which can be used for classification tasks. After the transformer blocks, these encoder models employ a pooling layer on the final hidden state of the [CLS] token as the final representation for the sequence which is then passed to a classifier layer for a classification or regression task. The pooler layer is a linear layer of size $d \times d$, where $d$ is the hidden dimension of the transformer with a tanh activation (in this case $d = 768$). A EuGen layer can be used as a drop-in replacement for this pooler layer. For this experiment, the base model is frozen and only the pooler and the classifier weights are tuned. We beat the SNNK layer (which has the same number of training parameters as ours) as well as the baseline model (Results for classifier and pooler are taken from [50] which has significantly more training parameters on **all** the 8 tasks.

## D.5  LLM Pre-training

In this section, we provide the implementation details of the LLM pre-training experiment. Our GPT-2 implementation is based on the open-source repository available https://github.com/karpathy/nanoGPT. We performed pre-training on 4 NVIDIA A6000 GPUs on OpenWebText dataset. In this experiment, we used 64 random features, 40 gradient accumulation steps, 0.1 weight decay, batch size of 36, and train the model for 50K iterations. We set the gradient clipping parameter to 1.0 for vanilla GPT-2 and set it to 0.0 for EUGen GPT-2. In Fig. 3, we present the results with 2 to 11 (out of 12) GPT layers replaced with EUGens, as we observed that replacing all layers affects the convergence.

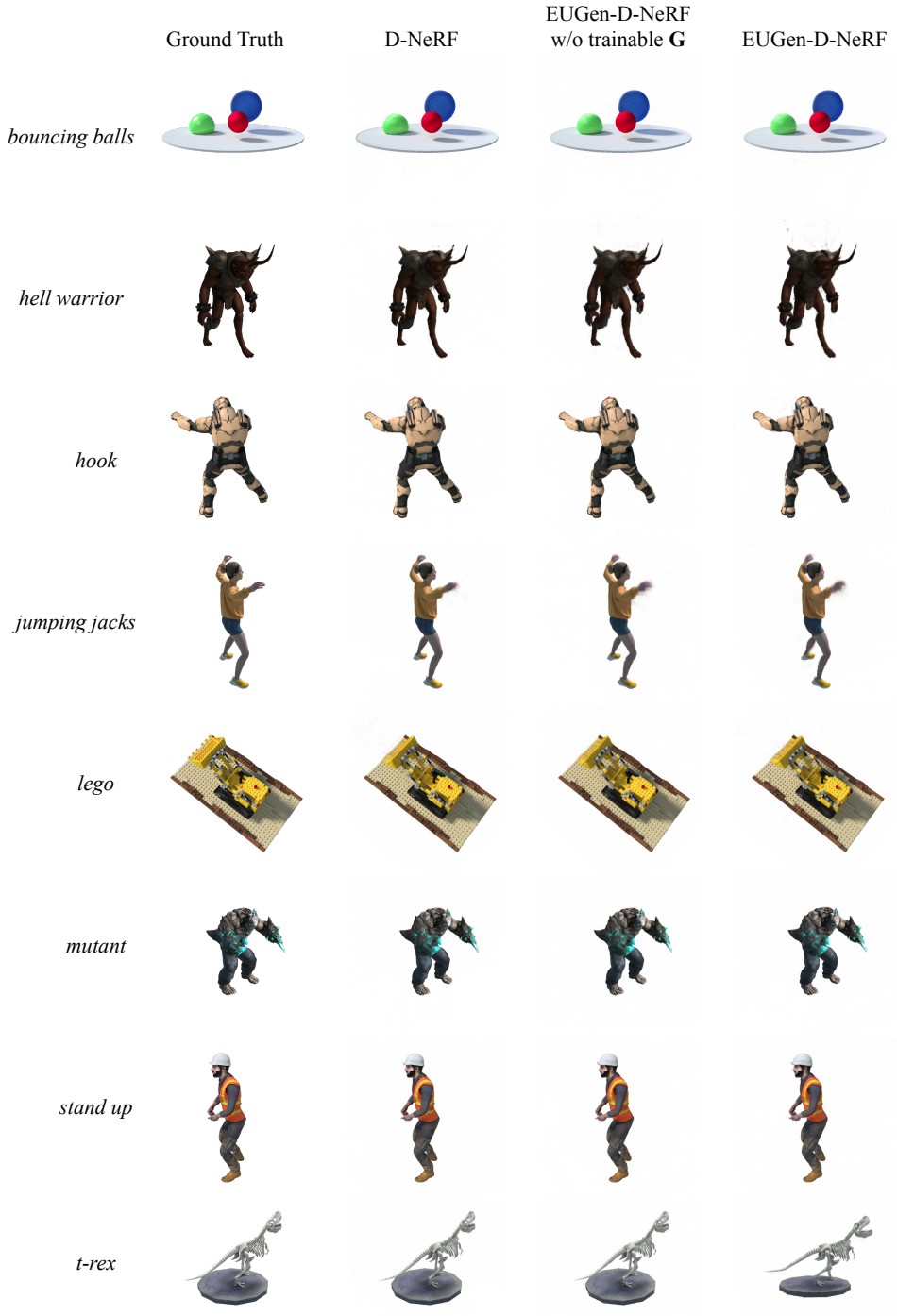

Figure 19: Rendered images from the D-NeRF dataset [71]. For each scene, the images from left to right show the Ground Truth, D-NeRF [71, 90], EUGen-D-NeRF without trainable **G**, and EUGen-D-NeRF. We observe that EUGens generate realistic renderings in all settings.

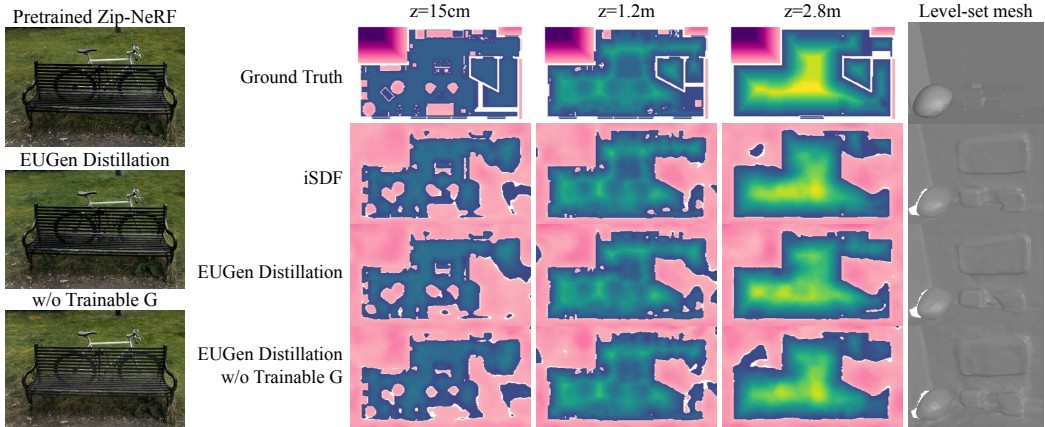

Figure 20: Visualization of EUGen distillation. (Left): Rendering of Zip-NeRF. (Right): Visualization of reconstructed SDFs. The last column shows the zoom-in rendering of zero level-set mesh.

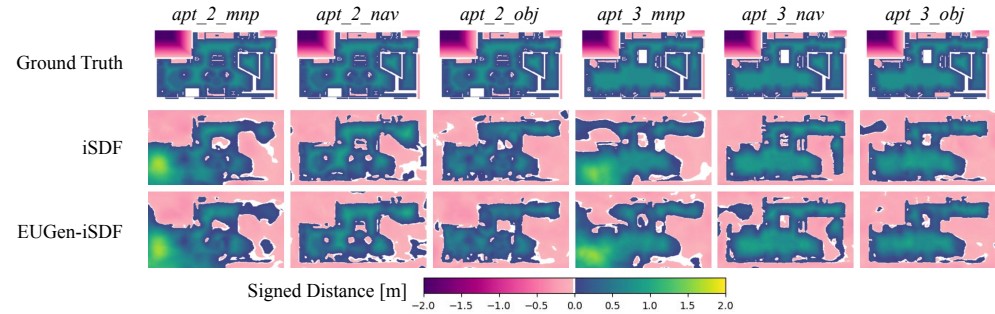

Figure 21: Reconstructed SDFs in iSDF and our EUGen-iSDF for six ReplicaCAD sequences.

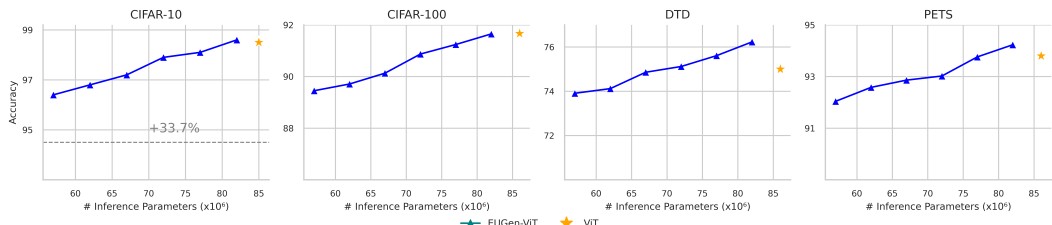

Figure 22: Accuracy vs Inference parameters trade-off plot for uptraining ViT by successively replacing MLP layers with the EUGen layers from the bottom. We show results for CIFAR-10, CIFAR-100, DTD, and Pets. We achieve almost **34**% reduction in parameters, while matching the performance of the baseline models in all cases.

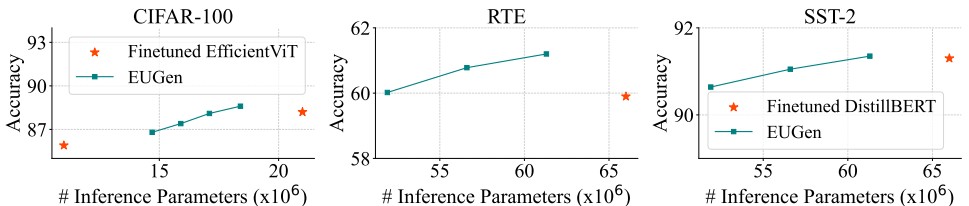

Figure 23: Uptraining experiments on other datasets and architectures. (Left) : Uptraining EfficientViT on CiFAR-100, (Middle and Right) : Uptraining DistilBERT on SST2 and RTE datasets. Our model can achieve similar performance as various baseline models while being significantly faster, thus becoming a complementary technique to distillation and other efficient architectures.

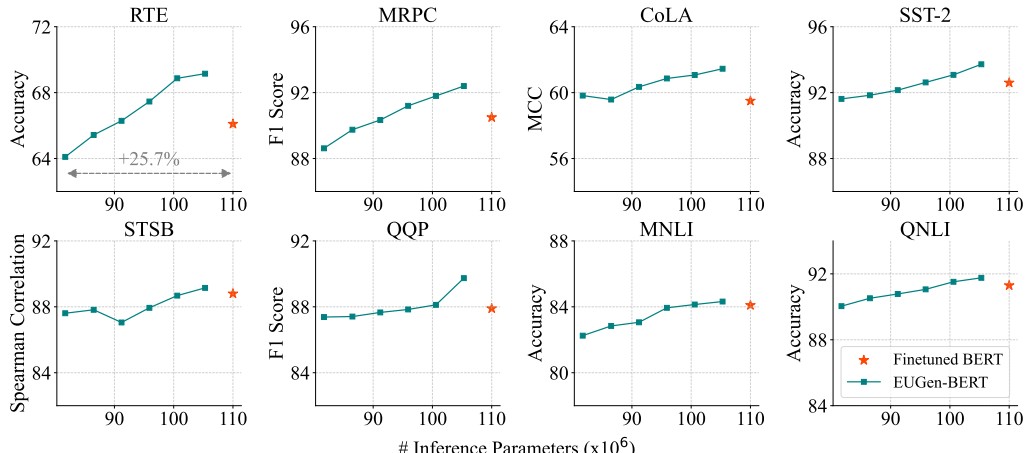

Figure 24: Detailed evaluation results of EUGens on the GLUE benchmark. MCC score is reported for CoLA, F1 score is reported for MRPC and QQP, Spearman correlation is reported for STSB, and accuracy scores are reported for the other tasks. EUGen matches the baseline performance while being almost **26**% smaller.

# E  Ablation Studies

In this section, we present additional experiments on **(a)** justifying the use of ReLU over Softplus, **(b)** Effect of the number of random features in various models, **(c)** justify the choice of using ORF in our EUGen layers, **(d)** justify the choice of training projection matrices **G** and **(e)** finally, show that introducing non-linearity is essential to the performance of EUGen layers

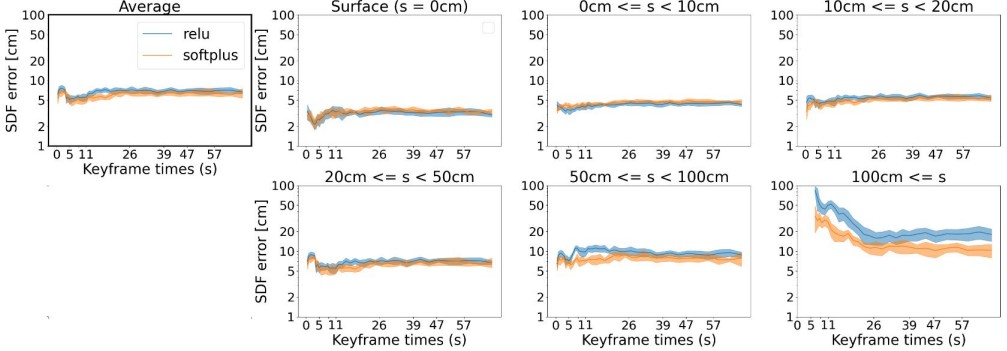

Figure 25: Visualization of the error when comparing ground truth and reconstructed SDF using various activation functions. The first column displays the average error. The subsequent rows illustrate errors across different ranges of ground truth SDF values. In robotics scenarios such as grasping or manipulation, achieving more precise reconstruction, particularly for lower SDF values as shown in the first row, is crucial.

## E.1  ReLU over Softplus in iSDF

As our goal is to explore how far we can accelerate using EUGen layers, we first investigate potential speed improvements in the baseline model before making modifications. We found that replacing Softplus with ReLU results in faster inference with minimal quality degradation. Specifically, this change causes a slight performance decrease for distances $s \geq 100$ cm (see Figure 25). However, since surface information is critical for robotics tasks such as manipulation or navigation, we exploit ReLU activation in our iSDF experiments both in baseline (iSDF) and EUGen-iSDF to prioritize network speed.

Table 8: Comparison between NeRF and our random feature mechanism on the *drums* dataset. The inference time refers to the approximate time for forward passes of each model to render a single 400x400 image.

| Method | # RF | Rendering Quality | | | Efficiency | |
| | | PSNR ↑ | SSIM ↑ | LPIPS ↓ | $t_{\text{infer}}$ (s) ↓ | Model Size (MB) ↓ |
|---|---|---|---|---|---|---|
| NeRF | - | 25.46 | 0.930 | 0.049 | 2.36 | 3.27 |
| EUGen-NeRF w/o trainable **G** | 256 | 24.92 | 0.922 | 0.056 | 1.79 | 2.27 |
| | 128 | 24.64 | 0.918 | 0.059 | 1.47 | 1.39 |
| | 64 | 24.39 | 0.913 | 0.064 | 1.32 | 0.96 |
| | 32 | 23.90 | 0.903 | 0.076 | 1.22 | 0.74 |
| | 16 | 23.17 | 0.886 | 0.098 | 1.19 | 0.63 |

## E.2  Effect of Number of Random Features in 3D Reconstruction Tasks

**Effect of Number of Random Features in EUGen-NeRF**. In this experiment, we investigate how the number of random features affects the quality of EUGen-NeRF. We evaluate three scenes: *hotdog*, *materials*, and *drums*, which represent easy, medium, and hard cases, respectively, based on rendering quality in Table 1. We test models with 16, 32, 64, 128, and 256 random features.

We present the qualitative and quantitative results of the number of random features in EUGen-NeRF in Figure 26 and Table 8. Using a small number of random features, like 16, results in approximately a two-fold acceleration compared to the baseline model.

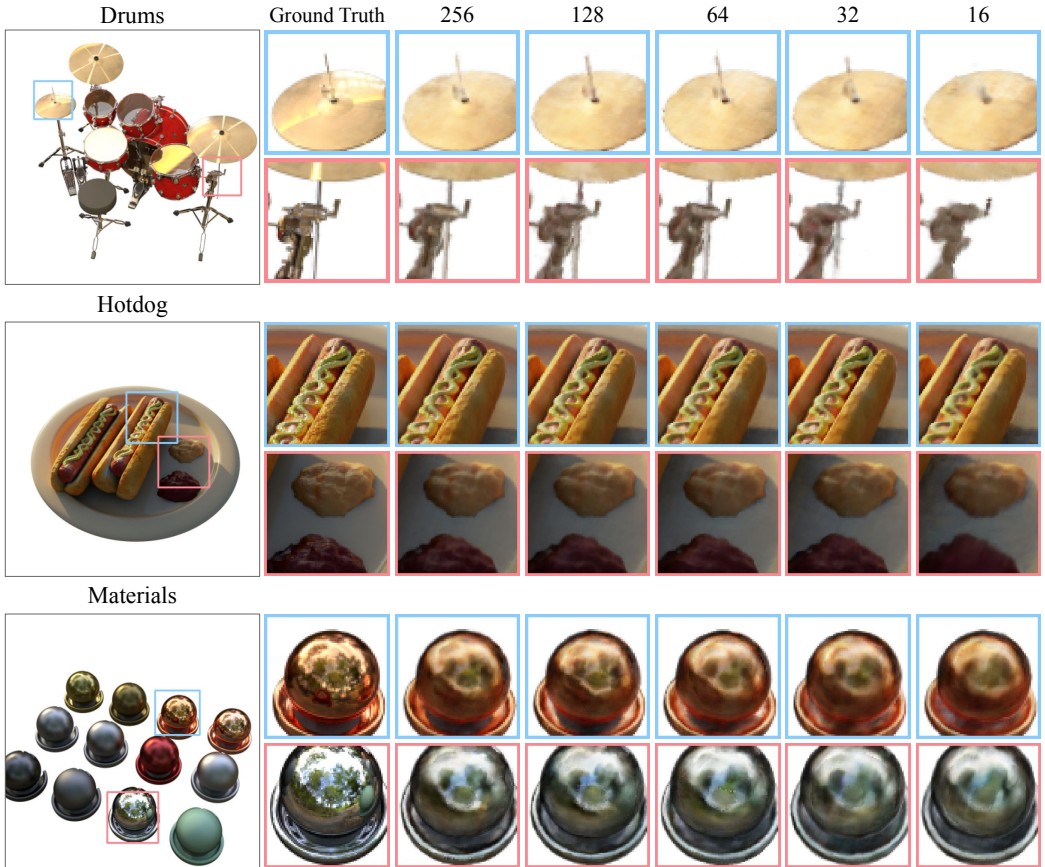

Figure 26: Comparison of rendering results with varying numbers of random features in EUGen-NeRF. The numbers at the top indicate the number of random features used for each specific EUGen-NeRF model. The ground truth image is on the left, while the right shows zoomed-in views of each reconstruction.

Finally, in an extreme setting, increasing the number of random features to match the baseline leads to improved results, demonstrating that EUGen-NeRF learns better representations than the baseline NeRF.

Table 9: Matching the baseline's parameter count by increasing random features leads to improved performance.

|  | PSNR ↑ | SSIM ↑ | LPIPS ↓ |
|---|---|---|---|
| baseline | 30.45 | 0.950 | 0.032 |
| EUGen (same # params) | **30.54** | 0.950 | **0.030** |

**Effect of Number of Random Features in Zip-NeRF Distillation**.

In Figure 27 and Table 10, we present an ablation study to illustrate how the number of random features affects the distillation process. The number of random features balances the trade-off between quality and inference speeds and storage costs.

**Effect on number of random features for iSDF experiments**. In this subsection, we show how the number of random features affect the results in our iSDF experiments. For this study, we use only the

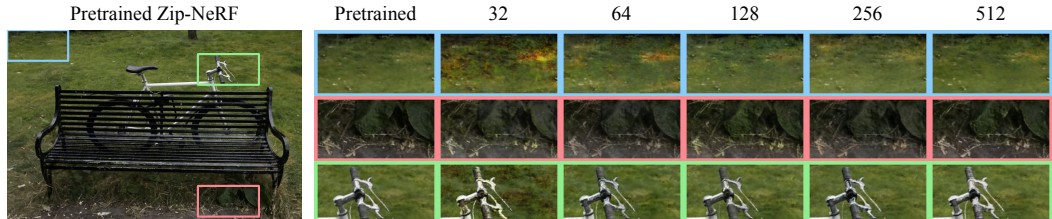

Figure 27: Comparison of rendering results with varying numbers of random features in Zip-NeRF distillation. The numbers at the top indicate the number of random features. Rendering from the target pretrained model is on the left, with zoomed-in views of each reconstruction on the right.

Table 10: PSNR of rendered images and inference time for pretrained Zip-NeRF and layer-wise distillation for our analytic variant using various number of random features on the *bicycle* dataset [4].

|  | Pretrained | $m = 32$ | $m = 64$ | $m = 128$ | $m = 256$ | $m = 512$ |
|---|---|---|---|---|---|---|
| PSNR ↑ | 23.66 | 20.72 | 21.06 | 21.55 | 21.72 | 22.06 |
| Inference time (s) ↓ | 4.03 | 3.76 | 3.78 | 3.86 | 4.06 | 4.43 |

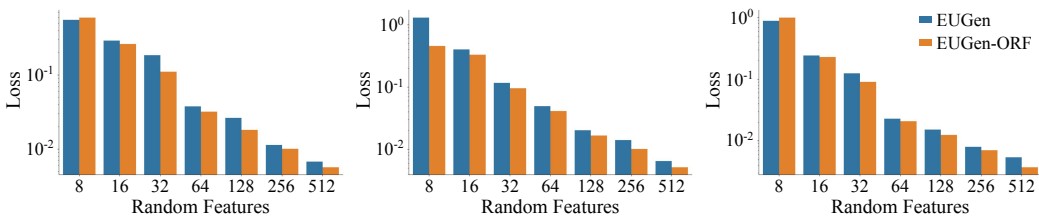

Figure 28: Comparing the approximation capability of our EUGen layer with orthogonal random features vs random features. (Left) : Approximating a ReLU-Linear layer. (Middle) : Approximating a Softplus-Linear layer. (Right) : Approximating a GELU-Linear layer.

non-trainable **G** variant. As before, we show that we can tradeoff accuracy vs speed by controlling the number of random features (Table 11). Qualitative results are shown in Figure 30.

### E.3 Orthogonal Random Features in EUGen

In this subsection, we show that ORFs in EUGen outperform random projections in approximating ReLU, Softplus, and GELU linear layers (Fig. 28).

### E.4 Trainable G improves performance

Recent works [19, 121] have shown that training the projection matrix **G** instead of using a fixed probability distribution can lead to improved results. We validate this hypothesis in the context of various 3D scene rendering tasks for NeRF, D-NeRF, and Zip-NeRF (Fig. 29).

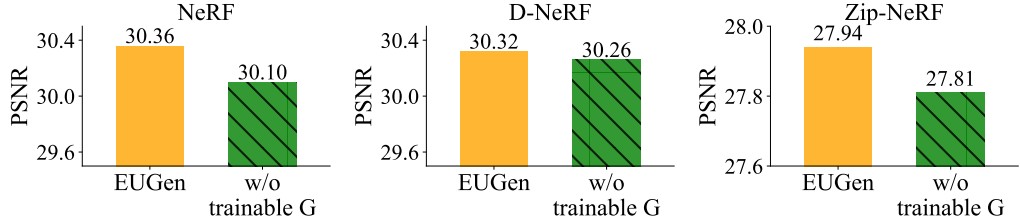

Figure 29: Trainable **G** improves performance for 3D scene rendering tasks for various architectures. (Left) : NeRF, (Middle) : D-NeRF, and (Right) : Zip-NERF

**Distillation Experiments with Trainable Projection Matrix**. In this subsection, we show that using a trainable **G** also improves the quality of the distilled models. However, in this case, optimizing the EUGen layer to mimic the outputs of an FFL in a pretrained INR can not be done analytically. So we use gradient descent to train this EUGen layer. Table 12 shows the results for various loss functions that can be used to compare the output of an EUGen layer with the target FFL. We observe that the Huber loss performs better than MSE as it is robust to outliers. Adding a trainable bias improves performance, but does not add any extra latency during inference. That is why the trainable **G** with bias and Huber loss is the preferred choice for **all** the distillation experiments.

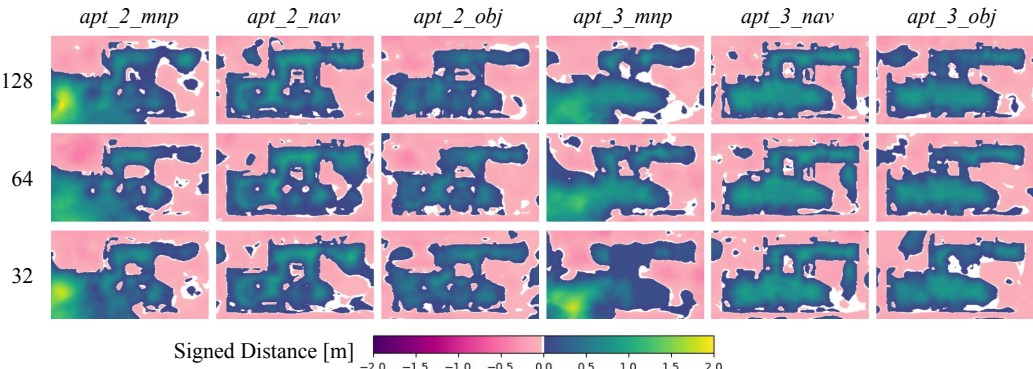

Figure 30: Reconstructed SDFs in iSDF for six ReplicaCAD scenes with varying numbers of random features. Each row corresponds to a different random feature count (indicated on the left) and follows the colormap shown at the bottom. Trainable **G** is not used for this experiment.

Table 11: Quantitative results of EUGen-iSDF. RF is the number of random features, $L1_{avg}$ is calculated across all SDF levels, and $L1_{surf}$ is for ground truth surface points. $t_{train}$ refers to iteration time (including backward pass), and $t_{infer}$ measures forward pass inference time. Results are averaged across 6 ReplicaCAD [86] sequences with 3 random seeds. For this ablation, the variant without trainable **G** is used.

| | Distance (cm) | | | Time (ms) | |
|---|---|---|---|---|---|
| # RFs | $L1_{avg}$ ($\downarrow$) | $L1_{surf}$ ($\downarrow$) | CD ($\downarrow$) | $t_{train}$ ($\downarrow$) | $t_{infer}$ ($\downarrow$) |
| 128 | 9.62 | 4.28 | 2.12 | 7.74 | 1.06 |
| 64 | 9.39 | 4.46 | 2.21 | 7.40 | 0.93 |
| 32 | 9.96 | 4.38 | 2.40 | 7.13 | 0.84 |

Table 12: Performance comparison of methods based on PSNR, SSIM, and LPIPS.

| Method | Loss Function | PSNR $\uparrow$ | SSIM $\uparrow$ | LPIPS $\downarrow$ |
|---|---|---|---|---|
| Baseline | N/A | 31.31 | 0.965 | 0.017 |
| Analytic (no backprop) | MSE | 28.54 | 0.948 | 0.027 |
| Trainable **G** | MSE | 31.07 | 0.964 | 0.017 |
| Trainable **G** | Huber loss | 31.11 | 0.964 | 0.017 |
| Trainable **G** + bias | MSE | 31.21 | 0.965 | 0.017 |
| Trainable **G** + bias | Huber loss | 31.25 | 0.965 | 0.017 |

## E.5 Effect of Nonlinearities

Non-linearity in EuGen arises from at least one of $\Phi$, $\Psi$, or $k \geq 2$. As discussed in Appendix B.3, a simplified variant reduces to a low-rank layer. We show that such low-rank layers consistently under-perform, underscoring the necessity of non-linearity.

Fig. 31 further illustrates this effect. We train various INR models with low rank layers having the same number of training parameters as our EUGen-models for a fair comparison. The performance

gap widens as more layers are replaced, with respective layer replacement counts of (3, 2, 2, 1) for (NeRF, iSDF, D-NeRF, Zip-NeRF).

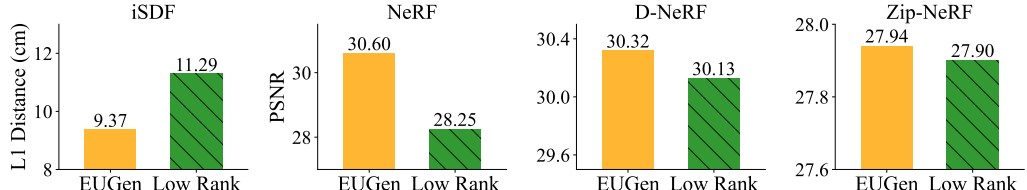

Figure 31: The importance of non-linearity in EUGen layers. Low rank variants underperform compared to EUGen models in various INRs.

We train a BERT model by replacing the expansion block of the final 3 Feed Forward Networks by low rank matrices with same number of training parameters as that of EUG-BERT. On 3 datasets, namely MRPC, COLA, and SST2 our method outperforms the low rank one (Fig 32).

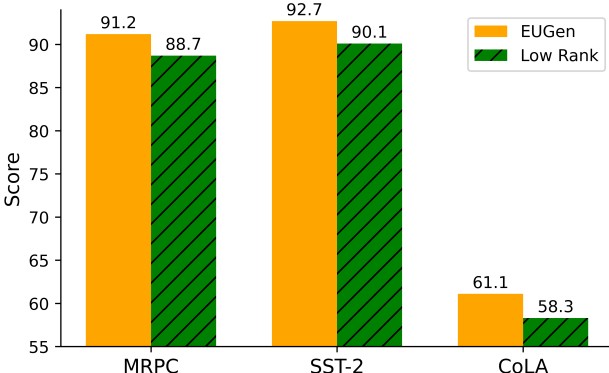

Figure 32: Comparsion of parameter matched low rank BERT with our EUGen-BERT. For all the 3 datasets, EUGen-BERT outperforms the low rank variant.

### E.6 Increasing the Polynomial degree

In this subsection, we will show that increasing the polynomial degree $k$ ($\leq 5$) can improve performance at the cost of inference speed. The experiments are run for GLUE language tasks (eight datasets) and CIFAR-100 from the vision-suite of tasks.

Table 13: Increasing the polynomial degree improves performance on the GLUE benchmark. MCC score is reported for CoLA, F1 score is reported for MRPC and QQP, Spearman correlation is reported for STSB, and accuracy scores are reported for the other tasks.

| Dataset | Baseline | EUGen, $k<5$ | EUGen, $k=5$ |
|---------|----------|--------------|--------------|
| RTE     | 57.5     | 61.7         | **62.1**     |
| MRPC    | 81.5     | 83.8         | **84.5**     |
| COLA    | 29.4     | **45.2**     | 44.4         |
| SST-2   | 84.9     | 86.8         | **87.4**     |
| STSB    | 78.1     | 82.8         | **82.9**     |
| QQP     | 72.0     | 72.8         | **73.4**     |
| MNLI    | 56.4     | 57.4         | **58.1**     |
| QNLI    | 74.5     | 75.5         | **75.6**     |

Average score on the GLUE benchmark while using lower-degree EUGen layers is 70.75 and that of its counterpart applying polynomials of degree $k = 5$ is **71.05**. The latter one is 3% slower in training and 2% slower in inference.

On CIFAR-100, EUGen layers of degree 3 achieve an accuracy of 89.53, while increasing the degree to 4 raises it to $\mathbf{89.81}$. Similar to GLUE results, we find that the higher-degree polynomial variant achieves better quality, at the price of the small speed loss (5% slower in training and 3% slower in inference).

We thus see here a classic speed-quality tradeoff, as well as improved quality for higher-degree polynomials, as expected.

# F   Broader Impact & Limitations

**Societal Impact**. This paper proposes an efficient neural network layer that lowers the computation footprint of fully connected feedforward layers (FFLs). The experiments with GPT-2, ViTs, NeRF, and iSDF illustrate how EUGens can lower the inference speed while remaining competitive with the baselines. We believe EUGens can enable the deployment of large-scale models in real-time systems. The application of EUGens will also help alleviate computational bottlenecks, thereby reducing the carbon footprint associated with running these models.

**Limitations**. Although EUGens yield significant improvements in inference speed and memory usage, more work is needed to comprehensively assess trade-offs while exploring the space of all possible $\Phi$ and $\Psi$. Furthermore, it is an open research question to design an efficient unbiased algorithm for approximating FFLs with general activation functions (beyond polynomials).

