# OpenReview forum: "EUGens: Efficient, Unified and General Dense Layers"
_NeurIPS.cc/2025/Conference — NeurIPS 2025 poster_

### Official Review · Reviewer_4fkV · 2025-06-29

**Clarity:** 3
**Significance:** 3
**Originality:** 3
**Rating:** 4
**Confidence:** 4

**Summary:**

The authors propose a new type of neural network layer called "EUGene", using random feature methods that speed up the inference time and reduce the training parameters of the network. The authors also provide a theoretical analysis of EUGen's estimation with the concentration bounds. The authors validate their approach by demonstrating how to integrate EUGen layers in popular architectures like tranformers and Neural Radiance Fields with experiments on LLM pre-training, vision transformers, and 3D image reconstruction.

**Questions:**

1) Figure 1: Why does the loss increase with more EUGene layers? Is this for the same reason as outlined in the experiment 2? If so, I would encourage the authors to address this in the earlier experiment than the one that comes later.
2) Figure 1: How does EUGene with more layers has fewer inference parameters (lower fraction of inference parameters)?
3) What are the important future work directions in the authors' views?
4) Why is theorem 3.1 relevant for neural network approximations? Polynomial activation functions are always explicitly excluded in all universal approximation theorems, because they lead to polynomial approximations - and these are extremely ill conditioned, i.e. do not really model classical neural network approximations. Also, as (l117) "in practice we use k <= 3", and l164 "k <= 2", so the polynomial approximation certainly is not "arbitrarily accurate" (for which k has to approach infinity). All theorems seem to be based on the polynomial activation assumption, which makes it very important to clarify this point. I may have missed an important argument for this assumption and am willing to raise the score if it can be explained properly. Simply stating that "Bernstein polynomials can approximate arbitrary continuous functions" is not enough, since k is very small.
5) l105: m is not defined, is it equal to l? Probably not, since l117 assumes m << min(l,d).

**Ethical Concerns:**

["NO or VERY MINOR ethics concerns only"]

**Final Justification:**

The paper presents impressive inference time speedups in numerical experiments, for a large range of network architectures. Although the theoretical setting in the paper misses the key points of the numerical algorithm, it still warrants publication in my view. If accepted, the authors should clearly indicate this mismatch between numerical experiments and theory in the main paper.

**Limitations:**

I would encourage the authors to make a separate section on limitations and future work and elaborate on the important points. This will help the reader. The limitations are mentioned, but are hidden in the details of the experiments.

**Paper Formatting Concerns:**

See questions; defining G properly is most important.

**Quality:**

3

**Strengths And Weaknesses:**

Strengths:
 * S1) Zero-shot integration is demonstrated on many important and commonly used architectures such as transformers and neural radiance fields, showing that the inference speed decreases without retraining with the classical back-propagation algorithm.
 * S2) Theoretical analysis including EUGene’s approximation results and concentration bounds is presented.
 * S3) Many architectures use feed-forward neural networks as building blocks and thus, this work can potentially impact many applications.
 * S4) Writing is clear and easy to follow.

Weaknesses:
 * W1) Important future work directions and limitations are not adequately detailed.
 * W2) (small weakness): notation in line 103 is confusing (w+, x+), because "+" is usually used to denote "pseudo inverse". Maybe "w_1" and "x_1" can be used to denote appending ones.
 * W3) l105: "the vectors" G are not defined. What are they? Trainable parameters, derived from W? Not clear at this point.
 * W4) The theoretical analysis is based entirely on the assumption that the modelled network layer has a polynomial activation function. this is excluded explicitly in universal approximation theorems (see questions).

---

> ### Author Rebuttal · Authors · 2025-07-30
>
> We would like to sincerely thank the Reviewer for valuable comments and feedback. We address all the comments below.
>
>
> > Questions about Figure 1:
>
> We believe the Reviewer is referring to Fig. 3 and 4. In our experiments, we keep the total number of layers in GPT-2 fixed (12 layers). The stated numbers L of layers refer to the numbers of layers replaced by EUGen layers. The increase in loss is expected as an increased number of EUGen layers corresponds to more approximation error in the network (we provided detailed results in the Appendix Fig. 17 & 18).
>
> > How does EUGene with more layers has fewer inference parameters?
>
> For EuGen the number of inference parameters is smaller and below we provide a brief explanation. A EuGen layer is of the form $f(W)g(x)$ and thus can be combined with a subsequent linear layer. Please see Fig 11 and the exact derivations in App. B1. In Transformers, an MLP block applies an up-projection using a weight matrix of shape $[4d, d]$, with a GeLU non-linearity followed by a down-projection via a matrix of shape $[d, 4d]$. Note that when we apply EuGens, we replace the up-projection + GeLU with an EuGen layer which has a shape $[4d, m]$, where m is the number of random features. This weight matrix of the EuGen layer can be combined with the down-projection matrix to create a single weight matrix $[d, m]$ and thus the number of the parameters in our compressed models scales as $O(d)$ as opposed to $O(d^2)$ of the baseline. This is also explained in l. 870-877 (App. D3).
>
> We will include all these clarifications in the paper.
>
> > vectors G and m:
>
> The number of random features is denoted by $m$. The $\mathbf{G}$’s are projection matrices with entries drawn randomly from a scalar Gaussian distribution, encoding corresponding $m$ Gaussian projections. We will make it clear in the updated draft.
>
> > notation in line 103 is confusing (w+, x+)
>
> Thanks for pointing this out. We will update the notation in the final version.
>
> > Future work directions and limitations:
>
> We believe we can improve EUGens via the following future research directions:
>
> * Currently EUGen provides unbiased estimation of the FFLs using polynomial activations. Extending this unbiasedness to arbitrary activation functions is an open problem (we already explicitly mention that limitation in Appendix F).
>
> * It would be interesting to see whether it is possible to apply polynomial function approximations using random features in both attention and MLP modules to maximize computational gains.
>
> * Finally, in the future work we also would like to explore efficient training strategies (and provide corresponding theoretical analysis) for models with all regular layers replaced with EUGen layers.
>
> We would like to note that such models are not polynomials neural networks in general, since in the most general definition of EuGens (Eq. 1), maps $\Phi$ and $\Psi$ explicitly depend on the norms of the input-vectors and weight-vectors respectively. Those models are more expressive than approximators of the regular FFLs with polynomial activations. We already successfully applied such general EUGen layers in this paper (look: comment in l.47-48, Sec. 4.4 and l.207 there).
>
> Appendix F currently includes our Broader Impact and Limitations section; we will expand this section to address point 2 and 3 (we already discuss point 1 in Appendix F).
>
>
> > Relevance of Theorem 3.1 for neural network approximations:
>
> We would like to sincerely thank the Reviewer for the question.
>
> * While pure polynomial-activation neural networks (PNNs) may have limited expressiveness unless the degree $k$ is very large, those results do not directly apply when polynomial-activation layers are mixed with non-polynomial activation layers. Our work uses hybrid architectures, which is a mixture of regular feedforward and EUGen layers. Our experiments show that our hybrid models, with a small degree $k$, achieve performance comparable to their standard counterparts while offering significant computational gains (see empirical evidence in Sec. 4.2 and 4.3). The theoretical analysis of the standard polynomial-activation NNs (PNNs) no longer holds. Thus our approximation theorems should be read in the context of the hybrid setting, where the corresponding layers are used, rather than pure PNNs.
>
> * Pure PNNs, despite their more limited expressiveness, are successfully used in several applications these days, e.g. (1) to replace deep convolution layers ([1]), (2) in dynamical system simulation, where their properties are particularly relevant for modeling ODE solutions and where they provide better accuracy and computational footprint that traditional numerical solvers ([2]), (3) in regression and classification tasks, where PNNs with additional tricks (such as the so-called product activations) turn out to be very strong learning models and are very relevant for Bayesian learning, as the authors explicitly admit, due to the corresponding closed-form calculations ([3]). Finally, in the recent work ([4]), lower bounds on the polynomial degree beyond which PNNs achieve maximum expressiveness are given via a rigorous theoretical analysis. Those theoretical results provide yet another evidence of the relevance of PNNs, even beyond hybrid architectures.
>
> * The above discussion regarding hybrid architectures and customized training procedures is analogous to the discussion on polynomial vs exponential functions in the attention setting. In that context, polynomial kernels (especially low-degree) were originally considered as not applicable due to their limited expressiveness (e.g. polynomial-capacity associated memory, instead of the exponential, as it is the case for the standard softmax-attention kernel). However low-degree polynomial kernels are nowadays used on the regular basis in attention modules of Transformers (see for instance: [5], where degree k=2 is applied or [6] where low-degree polynomials are used), providing comparable quality and significant reduction of the memory footprint. This is exactly due to new training procedures and hybrid strategies (e.g. up-training mechanisms, where regular Transformer is fine-tuned with some of the regular attention layers replaced by the polynomial ones).
>
> * Studying theoretical properties of PNNs is very important in the context of understanding regular deep NNs, as experts in the field directly admit. Because of that and the fact that their theoretical analysis is more tractable, several top-tier papers focus solely on analyzing their theoretical properties. A flagship paper is a manuscript of Kileel, Trager and Bruna ([7]), where the authors say: “...we believe polynomial networks can help us access a better understanding of deep nonlinear architectures, for which a precise theoretical analysis has been extremely difficult to obtain. Furthermore, polynomials can be used to approximate any continuous activation function over any compact support (Stone-Weierstrass theorem). For these reasons, developing a theory of deep polynomial networks is likely to pay dividends in building understanding of general neural networks…”. Our paper goes beyond theoretical analysis, by balancing theoretical results with strong empirical evidence (as the Reviewer mentions: “Zero-shot integration is demonstrated on many important and commonly used architectures such as transformers and neural radiance fields”). Because of that and the above comment, we do believe that the results on approximating polynomial-activation FFLs with random features (Theorems 3.1 - 3.4) are relevant and important, even outside of the scope of hybrid models.
>
> We completely agree with the Reviewer that that should be clarified and thus will provide the above clarification in the final version of the paper.
>
>
> [1] P-nets: Deep Polynomial Neural Networks; Grigorios G. Chrysos et al.  CVPR 2020
>
> [2] Polynomial Neural Networks and Taylor Maps for Dynamical Systems Simulation and Learning; Andrei Ivanov et al. 2019
>
> [3] Ladder Polynomial Neural Networks; Li-Ping Liu et al.  2021
>
> [4] Activation thresholds and expressiveness of polynomial neural networks; Bella Finkel et al., 2024
>
> [5] Simple linear attention language models balance the recall-throughput tradeoff; Simran Arora et al. ICML 2024
>
> [6]  PolySketchFormer: Fast Transformers via Sketching Polynomial Kernels; Praneeth Kacham et al. ICML 2024
>
> [7] On the Expressive Power of Deep Polynomial Neural Networks; Joe Kileel et al. NeurIPS 2019
>
>
>
> > Higher-degree polynomials for EUGens:
>
> To ground the discussion on higher-degree polynomials in EUGens, for the rebuttal we have conducted additional experiments, increasing the degree of the polynomial  up to k=5. The experiments were run for GLUE language tasks (8 datasets) and CIFAR-100 from the vision-suite of tasks.  The averaged results are presented below :
>
> GLUE :
>
> |baseline	|EUGen k<5 | EUGen, k=5|
> |-|-|-|
> |66.79	|70.75|	71.05 |
>
> Please find the full results in Reviewer DUMo’s response. We observe an improved quality for higher-degree polynomials using EUGens.
>
>
> CiFAR - 100 :
>
> |EuGen w/  k<4 | EuGen w/  k=4 |
> |-|-|
> |89.53 | 89.81 |
>
>
> For CIFAR-100 the conclusions are similar to those for GLUE tasks: the higher-degree polynomial variant achieves better quality, at the price of the small speed loss (5% slower in training and 3% slower in inference).
>
> To summarize, increasing degree helps slightly, but given the overall quality-speed tradeoff, we found the approach taken in the paper more optimal.

---

> > ### Comment · Reviewer_4fkV · 2025-08-06
> >
> > I appreciate the detailed answers to my questions.
> > The answer regarding theory by the authors means that most (if not all) computational results in the paper are not covered by the theory (where k is large, and norm(x) is not included - but it seems necessary for good performance in practice). This is unfortunate - but may warrant future work beyond the scope of this paper. I recommend to very clearly state this after the main theorem, and avoid to argue w.r.t. Bernstein polynomials. All other concerns are addressed adequately, I will raise my score accordingly.

---

> > > ### Author Response · Authors · 2025-08-06
> > > **Thank you note for Reviewer 4fkV**
> > >
> > > We would like to sincerely thank Reviewer for the new comments. We will make all suggested changes in the final version of the paper, as suggested by the Reviewer.
> > >
> > > Yours sincerely,
> > >
> > > The Authors

---

> ### Author Response · Authors · 2025-08-03
> **Feedback of Reviewer 4fkV**
>
> We would like to once more sincerely thank the Reviewer for feedback and very valuable comments. We believe that in the rebuttal we have addressed in depth all the questions. In particular:
>
> 1. We provided detailed discussion regarding connection with polynomial neural networks (PNNs) in the context of the presented theoretical results.
> 2. We provided detailed discussion regarding limitations and future work.
> 3. We provided details regarding EUGen scaling properties and provided detailed technical explanation why networks using more EUGen layers are characterized by fewer inference parameters.
>
> In addition, to complement our discussion on the connection with PNNs, we have run additional ablation studies on activation functions using higher-degree polynomials (for both: language and vision tasks).
>
> If the Reviewer has any additional questions, we will be very happy to answer them. Otherwise, we would like to sincerely ask the Reviewer to update the score accordingly.
>
> Yours sincerely,
>
> The Authors

---

> > ### Author Response · Authors · 2025-08-06
> > **comments on the rebuttal**
> >
> > Dear Reviewer 4fkV,
> >
> > We would like to once more sincerely thank you for the feedback and very valuable comments ! We addressed all the questions in the detailed rebuttal. We would like to once more reiterate our request to comment on the rebuttal and update the score accordingly since the end of the discussion period is approaching.
> >
> > Thank you very much !
> >
> > Yours sincerely,
> >
> > The Authors

---

### Official Review · Reviewer_QK92 · 2025-07-01

**Clarity:** 3
**Significance:** 3
**Originality:** 3
**Rating:** 5
**Confidence:** 3

**Summary:**

The authors propose a novel approximation to the dense layer that has improved compute complexity in terms of layer size. Extensive experiments show end-to-end speed up over fully connected layers.

**Questions:**

For the speed-up experiment: How are the standard linear and EUGen layer implemented?

**Ethical Concerns:**

["NO or VERY MINOR ethics concerns only"]

**Final Justification:**

The most relevant of my concerns were adequately addressed in the rebuttal, so I'm raising my score. Yes, this paper does not fully detail whether EUGENS are helpful for all kinds of neural networks, but I am sufficiently convinced they can be beneficial for some of them to recommend acceptance.

**Limitations:**

yes

**Paper Formatting Concerns:**

formatting is good.

**Quality:**

2

**Strengths And Weaknesses:**

## Strengths:
* Extensive experiments.

* Practically relevant problem.

* Easy to interpret trade-off plots (e.g. Fig. 3b)


## Weaknesses:
* No code is given, making verification of the author's claims unreasonably time-consuming.

* Line 87 states that the key distinguishing feature of EUGens over SNNK is that "[SNNK] cannot be applied to unbounded activation functions (e.g. ReLU)", but Fig. 2 shows results for SNNK on a ReLU network --> can the authors please explain this contradiction?

* The claim  that model size can be reduced seems to reference line 210, and Tab. 7 -- Here we see that the 30% reduction in parameters causes >0.5dB degradation in output quality. This is far from equal qualtiy. I suggest the authors relativize this claim and encourage them to remove it from the abstract without more detailed explanation.

* Despite the misleading claims in the abstract, this method seems to reduce compute only (not memory use; or if so, this has not been convincingly proven). In this context it would be appropriate to explain that commonly e.g. LLMs are memory rather than compute bound, limiting the utility of this approach.

---

> ### Author Rebuttal · Authors · 2025-07-30
>
> We are happy to hear that the Reviewer thinks that our paper presents extensive experiments, showcasing the capabilities of our method.
>
> > Code
>
> We have provided the pointers to anonymous repositories at the bottom of Page 24 (Appendix).
>
> > SNNK for ReLU:
>
> Thank you very much for this question. SNNKs indeed cannot unbiasedly approximate FFLs with unbounded activations, which is not the case for EuGens, with instantiations providing unbiased activations of the FFLs with arbitrary polynomial activations (via random feature mechanisms), as stated in Theorem 3.1. Thus EuGeNs can accurately and unbiasedly approximate polynomial-activation FFLs without any learning, simply by sampling matrices $G$ form the pre-defined distributions defined in Theorem 3.1 (explicit approximation). To summarize, when SNNKs replace FFLs with unbounded activations, explicitly they provide a replacement unrelated to ReLU-activations. During the process of training, SNNKs are capable of approximating layers with ReLU-activations (implicit approximation), however this approximation still will not be unbiased. We agree with the Reviewer that this requires clarification, and thus we will provide the above explanation in the final version of the paper.
>
> > Reduction in model size and impact in quality, memory footprint:
>
> We would like to thank the Reviewer for the question regarding memory footprint. As the Reviewer suggests, we will provide detailed quality vs memory savings trade-off early on in the paper and update the abstract accordingly. Thank you very much for the comment !
>
> Our EUGen-models are characterized by the smaller number of parameters than their regular counterparts and thus indeed do have a lower memory footprint. For example, by replacing top 6 FFLs with EUGen layers, we managed to drastically reduce the size of the ViT model from **346 Mb to 196.85 Mb**. Furthermore, we show memory footprint improvements of the NeRF models applying EUGens in Table 7. We see that models with a smaller number of random features have a smaller corresponding memory footprint. Additional experiments that we have conducted for the rebuttal on GPT2 show memory footprint compression from 1.4 GB to 1.2GB, via EUGens.
> Of course we are subject to the classic quality-compute tradeoff here, but since EUGen layers can be applied surgically to replace selected layers in regular models (all experiments we have conducted in the paper are for **hybrid architectures** with a combination of regular FFLs and EUGens), we can customize the extent of memory footprint reduction at the cost of some quality loss to the particular application.
>
> We conducted additional experiments by training the **24-layer ViT-L** model. By replacing up to 21 layers with our proposed EUGen modules and training from scratch, we achieve performance similar to ViT-L model across both the ImageNet and Places365 benchmarks. These lead to a **50.67%** reduction in inference parameters. The memory required also reduces from 1.2GB to 0.6GB.
>
>
> We will clarify it in the final version of the paper.
>
> > Implementation of standard linear and EuGen layers
>
> Thank you very much for the comment. The linear layer is implemented via nn.Linear and the EuGen layer is implemented, as described in Fig 1. In a nutshell, we train two small networks: one to transform the weights and the other to transform the inputs into a common latent space where the weights and inputs are fused via standard matrix multiplications. Please see the code for more details. We will clarify it in the final version of the paper.

---

> > ### Comment · Reviewer_QK92 · 2025-08-07
> >
> > Thank you for the further explanations. My most pressing concerns have been adequately addressed.

---

> > > ### Author Response · Authors · 2025-08-07
> > > **response to Reviewer QK92**
> > >
> > > We would like to once more thank Reviewer QK92 for all the comments and feedback. We are happy that
> > > Reviewer's most pressing concerns have been adequately addressed. We would like to sincerely ask the Reviewer to update the score accordingly. If the Reviewer has any remaining questions, we are very happy to answer them.
> > >
> > > Yours sincerely,
> > >
> > > The Authors

---

> ### Author Response · Authors · 2025-08-03
> **Feedback of Reviewer QK92**
>
> We would like to once more sincerely thank the Reviewer for feedback and very valuable comments. We believe that in the rebuttal we have addressed in depth all the questions. In particular:
>
> 1. Following Reviewer's comments, we have run additional large-scale experiments with **24**-layer ViT-L architectures. By replacing up to **21** layers with our proposed EUGen modules and training from scratch, we achieve performance at par with the regular ViT-L model across both the ImageNet and Places365 benchmarks. These lead to a **50.67%** reduction in inference parameters. These results show that EUGen scales to very deep architectures and can be used to replace **most** of the corresponding regular layers.
> 2. We clarified that the paper includes the code and provided corresponding reference.
> 3. We provided detailed discussion regarding memory footprint of our models.
> 4. We clarified the connection with SNNK models.
>
> If the Reviewer has any additional questions, we will be very happy to answer them. Otherwise, we would like to sincerely ask the Reviewer to update the score accordingly.
>
> Yours sincerely,
>
> The Authors

---

> > ### Author Response · Authors · 2025-08-06
> > **comments on the rebuttal**
> >
> > Dear Reviewer QK92,
> >
> > We would like to once more sincerely thank you for the feedback and very valuable comments ! We addressed all the questions in the detailed rebuttal. We would like to once more reiterate our request to comment on the rebuttal and update the score accordingly since the end of the discussion period is approaching.
> >
> > Thank you very much !
> >
> > Yours sincerely,
> >
> > The Authors

---

### Official Review · Reviewer_NaGy · 2025-07-02

**Clarity:** 3
**Significance:** 3
**Originality:** 3
**Rating:** 3
**Confidence:** 3

**Summary:**

This paper presents EUGens, a new class of dense layers designed to approximate standard feedforward layers (FFLs) using random features. The approach is backed by theoretical guarantees showing unbiased approximation for polynomial activations, and the layers can be plugged into existing architectures like Transformers and NeRFs to reduce computational cost. The authors also propose a lightweight distillation method that allows swapping in EUGens without needing full backpropagation, which is a practical feature for real-world deployment.

**Questions:**

See Weakness Section.

**Ethical Concerns:**

["NO or VERY MINOR ethics concerns only"]

**Final Justification:**

Based on the points raised in my comments on the rebuttal, I have decided to lower my score from 4 to 3. While I acknowledge that the paper is grounded in strong theoretical foundations — which is certainly a strength — I believe the experimental validation requires further work to fully substantiate the claims.

**Quality:**

3

**Strengths And Weaknesses:**

Strengths
- The theoretical contributions are solid and well-presented. EUGens are shown to be unbiased approximators of FFLs with polynomial activations, supported by detailed variance and concentration bounds.
- The method is architecture-agnostic and works in diverse domains—from NLP and vision to 3D reconstruction—showing versatility.
- The empirical results demonstrate improvements in inference speed and memory usage, often without substantial quality degradation.
- The distillation technique that allows layer-wise replacement without backprop is clever and could be very useful for adapting pretrained models efficiently.

Weaknesses
- One concern I have is that the experimental results don't clearly demonstrate EUGens as a superior alternative. Figures 3, 4, and 5 all suggest that EUGens offer a trade-off between model size and performance. But it’s hard to judge how favorable this trade-off really is, since reducing the number of inference parameters generally comes at the cost of accuracy. To show that EUGens are truly competitive replacements for existing FFLs, I think it would help to include an experiment where the EUGen model is scaled up to match the inference parameter count of the baseline—so that we can directly compare performance under equal capacity.

- Another point that left me puzzled is the use of 2, 4, 6, and 11 EUGen layers in the experiments. This doesn’t feel like a clean or scalable setup. In particular, Figure 4 shows that the 11-layer configuration performs significantly worse, which raises the question: is EUGen's benefit limited in deeper models? If 11 layers already hurt performance—and they probably still make up a small part of the network—it might mean that EUGens don’t scale that well in depth. I’d appreciate more clarity on why these particular layer counts were chosen, and whether EUGens are actually viable when used more extensively in deeper architectures.

- Additionally, while the paper focuses a lot on inference efficiency, it doesn't say much about training cost—especially when the Gi_j matrices are learned. If EUGens are supposed to be efficient alternatives, the cost of training them should be addressed too.

- Finally, the distillation method assumes you can store and access the input-output pairs of the layers you want to replace, which might not always be feasible for large models or datasets.

---

> ### Author Rebuttal · Authors · 2025-07-30
>
> We are happy to hear that the Reviewer thinks that theoretical contributions of the paper are solid and well-presented and that the method works in diverse domains - from NLP and vision to 3D reconstruction, showing versatility.
>
> > Trade-off between model size and performance for EUGen (an experiment where the EUGen model is scaled up to match the inference parameter count of the baseline):
>
> As requested by the Reviewer, we have added an experiment where we matched exactly the number of the parameters of the baseline, while training the variant applying EUGen layers. The task was NerF model training. The NerF variant leveraging EUGen layers turned out to be **more accurate** than the baseline in terms of PSNR (higher is better), and virtually indistinguishable on others (SSIM (higher is better) and LPIPS (lower is better)). We provide details below.
>
> |                                 | PSNR  | SSIM  | LPIPS |
> |--------------------|-------|-------|-------|
> | baseline                        | 30.45 | 0.950  | 0.032 |
> | Eugen (same param number) | **30.54** | 0.950 | **0.030** |
>
>
>
> > EUGen scaling with depth (how scalable EuGen’s setup is):
>
> Thank you very much for the question. We apologize for the confusion. Fig 17 and18 in the Appendix present the expanded versions of the results from Fig 3. These experiments were conducted using a ViT-B model, which has a total of 12 layers. Our experiments show that we can replace up to 11 layers by EuGens with minimal drop in performance and have a **40%** parameter reduction. The FFLs in Transformers contain most of the training parameters. The increase in loss is expected as an increased number of EUGen layers corresponds to more approximation error in the network. We will clarify it in the final version of the paper.
>
> Following Reviewer’s comment, we conducted additional experiments by training the 24-layer ViT-L model. By replacing up to 21 layers with our proposed EUGen modules and training from scratch, we achieved performance at par with the ViT-L model across both the ImageNet and Places365 benchmarks. These led to a 50.67% reduction in inference parameters without sacrificing accuracy. This shows that our method can also be successfully applied to deeper models.
>
>
> > Cost of training:
>
> The projection matrices $G_{ij}$ are very small (to be more specific, for all fine-tuning experiments with ViT and BERT, we use 32 projections and  for most 3D reconstruction tasks  we apply 64 projections). Finally,  they incur only negligible extra cost in training. We will clarify it in the final version of the paper. Please see: Fig. 7 and Table 4 for training times of the iSDF models. We also demonstrated that iSDF training time can be improved by 5% with EUGens. This clearly shows that matrices $G_{ij}$ do not incur any significant extra cost in training. Table 9 in the Appendix further shows the training time as a function of the number of random features m. Models with smaller m are characterized by shorter training time.
>
> > Distillation:
>
> In practice, performing distillation using EUGens is not expensive because of the following reasons:
>
> We can perform forward passes using the teacher model during training and access the input activations in a batch-wise fashion. This strategy prevents storing large amounts of data in an offline setup.
> For specific models like Zip-NeRF, even the above strategy can be expensive due to the large number of samples required per image. To mitigate this, we adopt several strategies: (1) we sample only 1/16 of the training images, (2) we render them at 1/8 of the original resolution, and (3) we randomly sub-sample 10% of the resulting input-output pairs. These choices significantly reduce the memory footprint and allow the distillation process to run efficiently within our hardware constraints.

---

> > ### Comment · Reviewer_NaGy · 2025-08-07
> >
> > Thank you for providing the experimental results and clarifications. The performance on the NeRF training task looks solid and has addressed my initial concern.
> >
> > That said, I find it somewhat unusual that the validation of a neural network architectural component is primarily conducted on NeRF, which is a relatively specialized application. It would strengthen the paper if the authors could include results on more standard benchmarks for vision transformers on core tasks in the revision.
> >
> > As for the trade-off evaluation, it is always a complex and tricky issue. The authors mentioned they conducted additional experiments by training the 24-layer ViT-L model, but the rebuttal does not directly show or substantiate the claimed no sacrifice in accuracy. I think such a claim should be supported more explicitly with numbers. I do consider this line of experimentation quite important. Without a depth-wise study, the decision of how many layers to replace becomes a highly sensitive and ambiguous hyperparameter, which could significantly affect real-world applicability.

---

> ### Author Response · Authors · 2025-08-03
> **Feedback of Reviewer NaGy**
>
> We would like to once more sincerely thank the Reviewer for feedback and very valuable comments. We believe that in the rebuttal we have addressed in depth all the questions. In particular:
>
> 1. As requested by the Reviewer, we have added an experiment where we **exactly matched** the number of the parameters of the baseline, while training the variant applying EUGen layers. The task was NerF model training. The NerF variant leveraging EUGen layers turned out to be **more accurate** than the baseline in terms of PSNR (higher is better), and virtually indistinguishable on others (SSIM (higher is better) and LPIPS (lower is better)).
> 2. As requested by the Reviewer, we have run additional experiments with deeper neural network architectures. To be more specific, we conducted additional experiments by training the **24**-layer ViT-L model. By replacing up to **21** layers with our proposed EUGen modules and training from scratch, we achieved performance at par with the regular ViT-L model across both the ImageNet and Places365 benchmarks. These led to a **50.67%** reduction in inference parameters without sacrificing accuracy. This shows that our method can also be successfully applied to deeper models and can be used to replace **most** of the corresponding regular layers.
> 3. We have provided detailed discussion on the distillation and the cost of training (explaining why the additional computational overhead is negligible).
>
>
> If the Reviewer has any additional questions, we will be very happy to answer them. Otherwise, we would like to sincerely ask the Reviewer to update the score accordingly.
>
> Yours sincerely,
>
> The Authors

---

> > ### Author Response · Authors · 2025-08-06
> > **comments on the rebuttal**
> >
> > Dear Reviewer NaGy,
> >
> > We would like to once more sincerely thank you for the feedback and very valuable comments ! We addressed all the questions in the detailed rebuttal. We would like to once more reiterate our request to comment on the rebuttal and update the score accordingly since the end of the discussion period is approaching.
> >
> > Thank you very much !
> >
> > Yours sincerely,
> >
> > The Authors

---

> ### Author Response · Authors · 2025-08-07
> **response to Reviewer NaGy**
>
> We would like to sincerely thank the Reviewer NaGy for the comment and address two remaining questions:
>
> **1. Validation on NeRF:**
>
> Thank you for the comment ! NeRF is one of the most natural targets to measure the performance of such methods as EUGen since NerF usually applies regular feedforward fully connected architectures. EUGen replaces FFLs in those architectures by their more computationally efficient variants. Thus any potential quality loss would be much easier to observe in those models rather than in attention-based architectures (where attention can adapt to work even with mis-functioning FFLs). NerF is also an excellent example of the feedforward fully connected model that is presently used in SOTA systems (which cannot be said about pure feedforward fully connected models applied for vision or language tasks).
>
> **2. Trade-off evaluation:**
>
> Thank you for the comment. While training ViT-L from scratch, the model with 21 regular layers replaced by EUGen layers provided 1%+ improvement, as compared to baseline, across different datasets. We do agree with the Reviewer that providing these details is important and we will do it also in the final version of the paper.
>
> We would like to sincerely ask the Reviewer whether there are any remaining questions that the Reviewer would like us to answer.
>
> Thank you very much !
>
> Best regards,
>
> The Authors

---

### Official Review · Reviewer_DUMo · 2025-07-05

**Clarity:** 3
**Significance:** 3
**Originality:** 3
**Rating:** 4
**Confidence:** 3

**Summary:**

This paper introduces EUGens (Efficient, Unified, and General dense layers), a new class of dense layers that generalize standard fully-connected feedforward layers (FFLs) by leveraging random features. EUGens aim to improve the efficiency of neural networks by approximating FFLs and reducing their inference complexity from quadratic to linear time. The method goes beyond traditional FFLs by incorporating direct dependence on input norms, which enhances their expressive power. Additionally, EUGens can approximate FFLs with arbitrary polynomial activation functions, offering significant reductions in parameters and computational overhead. The paper also presents a layer-wise knowledge transfer technique that bypasses backpropagation for more efficient adaptation of EUGens to pre-trained models. Empirically, integrating EUGens into architectures like Transformers and MLPs yields improvements in inference speed (up to 27%) and memory efficiency (up to 30%) across tasks such as image classification, language model pre-training, and 3D scene reconstruction.

**Questions:**

Trade-off Between Expressiveness and Efficiency: The introduction of input norm dependence is a key feature of EUGens. How does this affect the trade-off between expressiveness and computational cost in practice? Are there specific scenarios where this dependence may be detrimental?

Hyperparameter Sensitivity: How sensitive is the performance of EUGens to changes in hyperparameters, such as the number of random features (m) and polynomial order (k)? Can the method adapt dynamically to different network sizes and tasks?

**Ethical Concerns:**

["NO or VERY MINOR ethics concerns only"]

**Limitations:**

Yes.

**Paper Formatting Concerns:**

No major formatting issues were found.

**Quality:**

3

**Strengths And Weaknesses:**

Strengths:
Novel Contribution: The introduction of EUGens provides an efficient alternative to traditional FFLs by incorporating random features and input norm dependencies. This novel approach reduces computational complexity, making it well-suited for large-scale neural networks.

Theoretical Foundation: The paper provides a thorough theoretical analysis, including proofs and concentration bounds that demonstrate EUGens' capability to approximate FFLs with polynomial activation functions. This strengthens the validity of the proposed method.

Empirical Validation: Extensive experiments show that EUGens outperform previous techniques (e.g., low-rank approximations) in terms of approximation quality (MSE loss), speedup, and memory usage. The method is applied successfully across various architectures, including Transformers, MLPs, and NeRF (Neural Radiance Fields), showing significant real-world applicability.

Knowledge Transfer and Distillation: The introduction of a distillation framework for layer-wise knowledge transfer without retraining is a significant innovation, improving deployment efficiency, especially for pretrained models.

Scalability and Practical Relevance: The paper highlights EUGens' potential for deploying large-scale neural networks in real-time, resource-constrained environments, making them highly relevant for practical applications.

Weaknesses:
Limited Task Diversity: The experimental evaluation mainly focuses on image classification, language models, and 3D scene reconstruction. While these are essential tasks, the paper could benefit from evaluating the method on a broader range of tasks (e.g., NLP beyond GPT-2, or tasks with sequential or graph-based data).

Impact of Input Norms: The introduction of direct dependence on input norms enhances expressiveness but may lead to inefficiencies in certain applications. Further discussion on the trade-offs between performance and computational cost when using this feature could help clarify its limitations.

Effect of Hyperparameters: While the paper evaluates the performance of EUGens with various hyperparameters (e.g., number of random features, polynomial order), a more detailed analysis of the sensitivity to these hyperparameters would provide deeper insights into the method's robustness.

Computational Complexity of Large Models: Although the method improves inference speed, the evaluation does not fully explore the scaling of EUGens with respect to very large models (e.g., extremely deep Transformers or large-scale NeRF setups). The impact of increasing model size or network depth on the gains provided by EUGens remains unclear.

---

> ### Author Rebuttal · Authors · 2025-07-30
>
> We are happy to hear that the Reviewer thinks that the paper provides a thorough theoretical analysis and extensive experiments complementing it.
>
> > Task diversity:
>
> We would like to sincerely thank the Reviewer for the thoughtful comment. As the Reviewer notes, our paper already targets a broad range of applications, including image classification, language models, and 3D scene reconstruction. In particular, we have conducted extensive experiments on image classification (ImageNet, Places365, CIFAR-10, CIFAR-100, DTD, PETS), NLP tasks (language modeling on 36.8 billion tokens as well as text classification using the GLUE benchmark), and 3D reconstruction (including unbounded and dynamic scenes as well as depth estimation). We believe these experiments span a wide breadth of topics relevant to the machine learning community. Furthermore, we have demonstrated that our method is truly general, as it can be integrated into various architectures (GPT, BERT, ViT, NeRF, Siren, EfficientViT), including large-scale models such as ViT-L. Taken together, we feel this already provides a solid diversity of tasks and architectures to support our claims.
>
> > Impact of input norms / Trade-off Between Expressiveness and Efficiency:
>
> Thank you very much for an excellent question. As shown in Eq. 1, the norm operator is applied independently on the weight-vectors and inputs. Since those weight-vectors and input-vectors are processed independently via nonlinear transformations, only to be fused in the final dot-product computations of the EUGen layers, the computational complexity of the variants with norms turned on is the same as for the variants that do not apply norms, up to the negligible time-complexity norm computation. That computation is indeed negligible as compared to the overall computations since it requires only one dot-product calculation (per weight-vector or input-vector; namely the dot-product of the vector with itself), whereas projections with matrices G involve several dot-product calculations.
> We will clarify it in the final version of the paper.
>
> > Effect of hyperparameters:
>
> Thank you very much for a great comment. In the paper,  we have already conducted detailed ablations on the number of random features (m). In l. 239, we discuss the effect of the number of random features on the quality and the computational footprint of the model, with results provided in Fig. 9 (right). This is for the 3D reconstruction task. We have also conducted those ablations for ViTs (see: Fig. 4). For additional results, please see: section E2 for other 3D reconstruction tasks (Fig. 26, 27, 30 and Table 7, 8) and Fig 28 for synthetic tasks. Our results show that the number of random features provides a classic trade-off between a model's computational footprint and quality.
>
> Following Reviewer’s suggestion, for the rebuttal we have also conducted additional experiments, increasing the degree of the polynomial to k = 5. The experiments were run for GLUE language tasks (eight datasets) and CIFAR-100 from the vision-suite of tasks.  The results are presented below :
>
> GLUE :
>
> |Dataset	|baseline	|EUGen, k<5 | EUGen, k=5|
> |----------------|-----------|------|----------|
> |RTE	|	57.5	|61.7|	62.1 |
> |MRPC |	81.5|	83.8|	84.5 |
> |COLA	 |	29.4|	45.2|	44.4 |
> |SST-2	|	84.9|	86.8|	87.4 |
> |STSB	|	78.1|	82.8|	82.9 |
> |QQP |		72|	72.8|	73.4 |
> |MNLI	|	56.4|	57.4|	58.1 |
> |QNLI	|	74.5	|75.5|	75.6 |
>
> Average score for the deep NN using lower-degree EUGen layers is 70.75 and that of its counterpart applying polynomials of degree k=5 is 71.05. The latter one is 3% slower in training and 2% slower in inference. We thus see here a classic speed-quality tradeoff, as well as improved quality for higher-degree polynomials, as expected.
>
>
> CiFAR - 100 :
>
> |EuGen with k<4 | EuGen with k=4 |
> |---------------------------|--------------------------|
> |89.53 | 89.81 |
>
>
> For CIFAR-100 the conclusions are similar to those for GLUE tasks: the higher-degree polynomial variant achieves better quality, at the price of the small speed loss (5% slower in training and 3% slower in inference).
>
>
> > EUGens in deeper models:
>
> We would like to thank the Reviewer for an excellent question. Following Reviewer’s comment, we conducted additional experiments by training the **24-layer ViT-L model**. By replacing up to **21 layers** with our proposed EUGen modules and training from scratch, we achieve performance at par with the ViT-L model across both the ImageNet and Places365 benchmarks. These lead to a **50.67% reduction in inference parameters**.

---

> ### Author Response · Authors · 2025-08-03
> **Feedback of Reviewer DUMo**
>
> We would like to once more sincerely thank the Reviewer for feedback and very valuable comments. We believe that in the rebuttal we have addressed in depth all the questions. In particular:
>
> 1. Following Reviewer's comment, we have run additional ablation studies over the role of the hyperparameters, by quantifying the role of the degree of the polynomials defining activation functions (for both: language and vision tasks). We have also pointed to the (already included in the paper) exhaustive empirical analysis of the role of the number of random features on the downstream performance.
> 2. Following Reviewer's comment, we conducted additional experiments by training the **24**-layer ViT-L model. By replacing up to **21** layers with our proposed EUGen modules and training from scratch, we achieve performance at par with the regular ViT-L model across both the ImageNet and Places365 benchmarks. These lead to a **50.67%** reduction in inference parameters. These results show that EUGen scales to very deep architectures and can be used to replace most of the corresponding regular layers.
> 3. We have provided detailed discussion on the impact of  inputs' norms (if directly included).
>
> If the Reviewer has any additional questions, we will be very happy to answer them. Otherwise, we would like to sincerely ask the Reviewer to update the score accordingly.
>
> Yours sincerely,
>
> The Authors

---

> > ### Author Response · Authors · 2025-08-06
> > **comments on the rebuttal**
> >
> > Dear Reviewer DUMo,
> >
> > We would like to once more sincerely thank you for the feedback and very valuable comments ! We addressed all the questions in the detailed rebuttal. We would like to once more reiterate our request to comment on the rebuttal and update the score accordingly since the end of the discussion period is approaching.
> >
> > Thank you very much !
> >
> > Yours sincerely,
> >
> > The Authors

---

> > > ### Author Response · Authors · 2025-08-07
> > > **Comments of Reviewer DUMo on the rebuttal**
> > >
> > > Dear Reviewer DUMo,
> > >
> > > We would like to once more sincerely thank Reviewer DUMo for feedback and very valuable comments on the paper and sincerely ask to comment on the rebuttal. If the Reviewer has any additional questions, we are very happy to address them. Otherwise, we would like to sincerely ask the Reviewer to update the score accordingly.
> > >
> > > Yours sincerely,
> > >
> > > The Authors

---

> ### Author Response · Authors · 2025-08-08
> **sincere request for a feedback on the rebuttal from Reviewer DUMo**
>
> We would like to thank the Reviewer for the initial feedback on the paper and very insightful comments. We would like to apologize for taking Reviewer's time, but we would like to once more sincerely ask Reviewer to comment on our rebuttal. We do believe that we have thoroughly addressed all Reviewer's comments. We have also conducted extensive additional experiments, following Reviewer's suggestions. Those involve in particular: (1) additional ablation studies over models' hyperparameters, as well as: (2) large-scale experiments with the ViT-L models. All those experiments support our original claims.
>
> The end of the extended discussion period is approaching and we would like to make sure that we can address all the remaining concerns (if any) or run any additional experiments (if needed). Thus the feedback from the Reviewer is very important for us. If the Reviewer does not have any additional questions, we would like to sincerely ask the Reviewer to update the score accordingly.
>
> Thank you very much !
>
> Yours sincerely,
>
> The Authors

---

### Note · Authors · 2025-08-12

Dear Area Chairs,

Thank you very much for your time and consideration. Our paper introduces a new class of the neural network layers, called *EUGen* layers. EUGens are of linear time complexity, with instantiations providing in particular: (1) the first unbiased estimation of the feed-forward layers with polynomial activations and: (2) a biased extension (with bias controlled by the complexity of the EUGen layer) for any smooth activation. This is, to our knowledge, the first such result.

The method is validated through extensive experiments spanning: image classification (ImageNet, Places365, CIFAR-10/100, DTD, PETS), NLP (GLUE benchmark, language modeling on 36.8B-tokens), and 3D reconstruction. EUGens show strong performance across various architectures including ViT, BERT, GPT, NeRF, iSDF, and EfficientViT. Reviewers have praised both the originality and the breadth of our results.

We have addressed all reviewers’ requests during the rebuttal, in particular by adding new ablation studies, experiments with deeper models such as ViT-L (showcasing up to **50%** reduction in inference parameters), and detailed clarifications of our theoretical results to Reviewer 4fkV as suggested.

---

### Decision · Program_Chairs · 2025-09-17

**Decision:**

Accept (poster)

**Comment:**

The reviewers acknowledged the relevance and novelty of the proposed method, its theoretical justification, and its effectiveness in terms of speedup and memory requirements. Nevertheless, they raised some concerns about the limited diversity of the tasks explored, the tradeoff between model size and accuracy, and the influence of some hyperparameters. Generally speaking, the authors' feedback convincingly addressed the reviewers concerns, with the exception of NaGy, who downgraded their score to a Borderline Reject. While Reviewer NaGy acknowledges the methodological contribution of the work, they still have some concerns about the empirical evaluation. However, the AC agrees with the other reviewers that the current set of experiments, complemented by those provided in the rebuttal, is sufficient to warrant acceptance. The authors are nonetheless strongly encourages to incorporate their feedback in the final version of the paper.